# Employee education, labor protection intensity and auditor risk perception

Xiaotian Shen[1], Anni Wu[2], Yi Ding[3], Qian Sun[4], Mengge Liu[1] *

1 School of Economics and Management, Beijing Jiaotong University, Beijing, China, 2 School of Business, Wuhan Huaxia University of Technology, Wuhan, China, 3 Business School, Renmin University of China, Beijing, China, 4 School of Economics and Management, Nanjing University of Science and Technology, Nanjing, China

* 22120679@bjtu.edu.cn

**Data Availability Statement:** Data Availability: The data used in this study are third-party data owned by CSMAR (China Stock Market & Accounting Research Database), therefore, the authors have no right to share the data. Interested persons can

## Abstract

Prior literature finds senior executives can influence auditor decision making. However, few studies have discussed the impact of employee's personal characteristics. Our research aims to fill the above research gaps by examining the impact of employee level education on audit costs. Taking A-share listed companies in Shanghai and Shenzhen from 2006 to 2021 as the research object, this paper examines the impact of employee education on audit fees. It is found that highly educated employees can effectively reduce the audit fees borne by the company, but the implementation of the Labor Protection Law weakens this inhibitory effect. In the case of low marketization level and weak Confucian culture intensity, employee education level has a more significant inhibitory effect on audit fees of listed companies. This study provides a basis for empirical research on the impact of employee attributes on auditor decision making, provides a new research perspective on the impact of labor protection law at the corporate micro level, and enriches the theoretical research on corporate governance rooted in traditional Chinese culture. We contribute to the practice that implications for evaluating the effectiveness of adopting labor protection.

## Introduction

Economic complexity in China has been shown to significantly impact the country's ecological footprint [1]. The nation's strategy of rejuvenation through science, education, and the cultivation of qualified personnel has notably elevated the education level of the Chinese population. The resulting demographic quality dividend is gradually compensating for the diminishing demographic quantity dividend, offering enduring support for sustained economic growth [2]. Demographic quality is closely linked to the stable economic growth and the overarching rejuvenation of the nation. As such, it has become a shared concern among enterprises, government entities, and the academic community. Consequently, the level of education has emerged as a crucial indicator for measuring the human capital within enterprises [3]. Rajan and Zingales [4] formalized the human capital theory of corporate governance, asserting that the governance challenge is no longer confined to the upper echelons of the corporate power structure but must extend to all employee levels. Presently, studies

contact CSMAR for the data (see https://data. csmar.com/ for more details, contact via email: service@csmar.com, tel: 400-639-8883). The authors confirm that they did not have any special access or privileges to the data that other researchers would not have.

**Funding:** The author(s) received no specific funding for this work.

**Competing interests:** The authors have declared that no competing interests exist.

predominantly concentrate on the impact of management education characteristics, including enterprise investment behavior [5], diversified decision-making [6], market participation [7], determination of subjective social status [8], and entrepreneurship quality [9]. While some research, based on questionnaire surveys, investigates the influence of all employees' education level characteristics on wage levels [10] and total profits [11], limited attention has been directed towards exploring whether the characteristics of all employees impact corporate governance and auditors' decisions. Audit fees, serving as a direct reflection of the auditor's risk estimation for the auditee, convey crucial signals to accounting information users. The study of audit fees is paramount for assessing audit market competitiveness and probing issues related to contracting and independence in the audit process [12]. Thus, a critical question arises: Do auditors believe that the education level of all employees diminishes audit risk by enhancing the quality of financial statements, leading to reduced audit fees? Additionally, how does the heightened intensity of labor protection mandated by the Labor Protection Law either foster or hinder the influence of highly educated employees on audit fees?

Based on this, our research aims to fill the above research gaps by examining the impact of employee level education on audit costs. This paper uses manually collected data on employee education level of Shanghai and Shenzhen A-share listed companies from 2006 to 2021 to measure employee education level, and tests whether the implementation of the Labor Protection Law can strengthen or inhibit the effect of highly educated employees on enterprise audit fees. Our findings reveal that companies with higher employee education levels tend to have lower audit fees compared to those with lower education levels. Moreover, we observe that labor protection significantly diminishes the inhibitory effect of highly educated employees on audit fees. Through a series of robustness tests, these findings persist. Further analysis indicates that the inhibitory impact of employee education on audit expenses is more pronounced in companies operating in less marketized environments and with weaker Confucian cultural influences.

Compared with previous studies, the contribution of this paper is as follows: Firstly, this paper expands the research scope of the impact of employee attributes on auditor decision making. Past research mainly focuses on the audit background of the executive [13], the accent of executives [14], background characteristics of executive management team [15], behavioral integrity of CEO [16] for auditor decision-making. This paper's significance lies in its use of manually collected data on employees' education levels, providing an accurate depiction of the educational qualifications of all employees within an enterprise. It affirms that the personal characteristics of employees beyond senior executives can indeed exert an influence on auditor decision-making.

Secondly, from the perspective of employees, this paper directly studies the effect of the implementation of the Labor Protection Law on the individual perspective of employees. Existing studies mainly explore the impact of the implementation of the Labor Protection Law on investment efficiency [17] and business elasticity [18] from the enterprise level, and economic growth from the region level [19]. This paper introduces a novel research perspective from the employees' standpoint, shedding light on the impact of the implementation of the Labor Protection Law at the micro-level of enterprises.

Finally, our study is confined to a single country (China), we acknowledge our findings may not generalize to other regions due to institutional differences [20]. Therefore, it is possible that the introduction of labor protection does not have the same effect on the auditing process in other countries. Unfortunately, we are unable to test our hypotheses in other countries because the required employee' education level data are unavailable. Even so, our study has implications for regions where labor protection is being considered for adoption or have been introduced. Meanwhile, when assessing the effectiveness of implementing labor protection

measures, regulators should place increased emphasis on the perception of audit risk. Where a partner of an audit project or a signed certified public accountant has actually undertaken the audit business of the same state-owned enterprise or listed company for a cumulative period of five years, he shall not participate in the audit business of the state-owned enterprise or listed company for five consecutive years thereafter.

## Literature review and theoretical analysis

### Literature review

Audit fees are the combined outcome of bargaining between auditors and clients, reflecting the auditor's requirement for risk compensation for audit risks, the auditor's corresponding investment and effort in ensuring the reliability of financial reports, and the firm's brand and bargaining power [12]. Specifically, based on the theory of "deep pocket", the auditor has certain insurance liability for the reliability of the client's financial reports. The higher the audit risk, the greater the risk of litigation, compensation, administrative penalties and damage to the auditor's brand reputation. Auditors will demand higher audit fees as compensation for risk [21,22]. Regardless of the substance of these allegations, firms subject to whistleblowing allegations have significantly higher audit fees [23]; On the other hand, the larger the client size and the higher the business complexity, the more resources need to be invested in the audit process, which will lead to the increase of audit fees [24]; Finally, the stronger the firm's bargaining power and the better its brand image are, the higher the audit fees it will generally charge [25]. It shows that the investment and effort of the auditor and the audit risk that the auditor needs to bear are the important determinants of audit fees. Although some studies have shown that audit fees are influenced by other factors such as the macro-systemic risks from the crisis [26], the extent of derivative hedging by clients [27] involvement of component auditors in multi-national enterprise [28], the district of a firm's headquarters [29], whether client firms rely on principles-based standards [30], shareholder activism [31]. The above factors still affect audit fees by influencing the auditor's investment and effort and the audit risk that the auditor needs to bear.

The level of audit risk faced by auditors in the execution of financial statement audit depends on the level of material misstatement risk and inspection risk, among which material misstatement risk can be further divided into inherent risk and control risk at the recognition level. When the level of material misstatement risk is high, auditors will choose to expand the scope of audit, obtain more audit evidence and other means to reduce the inspection risk, which will lead to the increase of audit resources investment, and ultimately the increase of audit fees. On the other hand, although the auditors will make efforts to reduce the inspection risk, the higher risk of material misstatement makes the auditors still face higher audit risk to a large extent, and the auditors must demand more premium to obtain risk compensation, which will also be reflected in the increase of audit fee level.

At present, the research on the influencing factors of audit fees mainly focuses on the characteristics of accounting firms, executives of listed companies, and auditors' personal characteristics. In characteristics of accounting firm, audit fees increase as time passes after implementing additional education requirements to enter the accounting profession [32]. The existing research shows that the law responsibility of the accounting firm [33], accounting firm organization form [34], transformation of accounting firms [35], the conversion of the accounting firms [36], significantly influence the audit fees of listed companies. In characteristics of executives of listed companies, relevant research has shown that managers ability [37], executive audit background [13], the chief executive rights [38], executive academic experience [39], CEO's duality [40] have a significant impact on audit fees of listed companies. In terms of

auditor personal characteristics, the auditors' personal characteristics may serve as a signal of the level of care that will be exercised during the audit process [41]. The existing studies have found that auditor industry expertise [42], auditor punishment [43] are significantly related to audit fees of listed companies. There are also some studies have discussed the impact of a portion of employee education on audit costs, such as audit fees increase as time passes after implementing additional education requirements to enter the accounting profession [32]. However, few studies have focused on whether the characteristics of all employees in a company affect corporate governance and auditor decision-making. Therefore, this paper uses manually collected data on employee education to accurately characterize the education level of all employees in order to confirm the influence of personal characteristics of employees other than senior executives on auditor decisions. Few studies find that firms with better ESG performance found to have less debt financing and easier access to equity capital from stock markets. Meanwhile, ESG has a significantly negative effect on a firm's financial performance [44] and dividend payouts [45]. Intense or prolonged sunlight exposure in the environment results in reduced discounts offered by investors [46]. However, the results did not show a significant impact of audit quality on this relationship [47].

Compare audit quality with in developed markets, some of China's institutions still need to be perfected. Audit quality concerns about involving component auditors are evident when subsidiaries are located in countries with weak institutions, and that there is overreliance on network auditors regardless of institutional environment [28]. Furthermore, China's tremendous and rapid economic and societal change places major pressure on management teams to effectively respond swiftly by taking more strategic actions [48]. On the other hand, prior research finds an increase in audit fees after IFRS adoption by European and Australian firms [49,50]. China also applies IFRS that is more principles-based than the US GAAP. Audit risk and audit fees are lower when client firms rely more on principles-based [51]. Meanwhile, the empirical studies in labor protection have been focused on developed countries [52,53], overlooking developing countries that the introduction of labor protection laws is more important to understand the impact of labor protection in emerging markets.

## Theoretical analysis

**Employee education level and audit fees.**   Previous studies have shown that firms with a high-quality workforce exhibit higher accruals quality, fewer internal control violations [54], fewer internal control deficiencies [55] and fewer financial statement restatements [56]; In terms of voluntary information disclosure, higher employee education level is associated with more frequent, timely, accurate and precise earnings forecasts [54].

Specifically, employees with higher education level generally improve the quality of financial reporting in the following two ways. First, employees with higher education level have comparative advantages in learning and using new technologies [57], and these employees can provide higher quality information for the accounting system. In other words, employees with higher education level can reduce the unintentional mistakes made in the process of collecting and generating data. The financial report generated by this part of information processing will have higher quality; Secondly, employees with higher education level are more aware of supervision [58], are associated with more corporate social responsibility engagement [59] and are more likely to realize the problem when the transaction is abnormal or even fraudulent, so as to report the information to the management before the transaction becomes a more serious misstatement. Call et al. found that senior executives will grant more stock rights to ordinary employees during the period of financial report fraud to prevent them from whistleblowing [60]. The level of education is also likely to increase the higher level of earnings manipulation

so that decreases quality of financial reporting. However, based on existing research, we consider that the results depend on the net effect of these two impacts, with the one that improves the quality of financial reporting having a greater impact.

Audit fees are expected to be high when auditors judge financial reporting as low quality [61]. And if the likelihood of financial statement manipulation high, auditors would expend more cost verifying unusually high accruals or inspecting high-risk accounts [62]. On the contrary, superior reporting quality lowers audit risk and the need for greater audit investments, such as auditors charge family firms less [63]. Excellent financial reporting quality both helps to ease the company internal and external information asymmetry problem [64], also helps to improve the auditor information provided to the customer's trust degree, which reduces the material misstatement risk assessment level and audit risk, and further characterized by lower audit fees [65].

Based on this, this paper proposes the following hypothesis:

**H1:** Employees with high educational background can effectively reduce the audit fees borne by the company.

**Labor protection, employee education level and audit fees.** When the Labor Contract Law is enacted in China, it is guided by the idea of giving employees more rights and protecting their legitimate interests [66]. Enterprises will be severely punished if they fire employees illegally, do not sign formal contracts with employees, and do not pay relevant expenses in full for employees. Therefore, after the implementation of the Labor Protection Law, firms adjusted their labor demand and raised the capital intensity of production [67], the cost of firing employees will be significantly increased, and employees will be more likely to obtain long-term and stable positions. The impact on labor protection intensity on the relationship between employee education and audit fees could go either way. For example, when labor protection intensity increases, lessens firm-level uncertainties, significantly reduces firms' business risk and accrual-based earnings [68]. When exploring the impact of employee education level on audit fees, this paper believes that labor protection mainly affects the inhibitory effect of employee education level on audit fees through the following two ways: first, the shirker effect. Under the protection of the Labor Protection Law, employees with higher education level are equally less likely to be fired by the enterprise for unintentional mistakes made in the process of data collection and data generation, which will promote the effect of protecting shirkers. Improvements in labor protection cause over-employment [69]. Based on this effect, employees with higher education who are more conscientious and less likely to make unintentional mistakes in their work will be lazy and slack off to a certain extent, thus failing to effectively improve the accuracy of information and the quality of financial statements, and thus failing to effectively reduce the audit fees borne by the company. The study of Ichino and Riphahn show that the improvement of labor protection intensity will significantly increase the slack degree of Italian white-collar workers at work [70]. Belot et al. also found that the improvement of labor protection intensity will protect the rights of shirkers, thus having a negative impact on employees' enthusiasm and work efficiency [71]. Second, the effect of organizational equity. Under the protection of the Labor Protection Low, less-educated workers will not be punished when they fall to proactively report problems when transactions are abnormal or even fraudulent. This will make highly educated employees after weigh the relationship between input and output ratio of injustice. And then affect the highly educated staff's work enthusiasm and work efficiency. Furthermore, the above impacts will not effectively improve the quality of financial statements of highly educated employees, and thus to effectively reduce the audit fees borne by the company.

Based on this, this paper puts forward the following hypothesis:

**H2:** Labor protection significantly weakens the inhibitory effect of highly educated employees on audit fees.

## Research design

### Sample selection and data sources

In this paper, A-share listed companies from 2006 to 2021 are selected as primary samples, and the research samples are screened as follows: (1) financial and insurance listed companies are excluded; (2) ST and *ST (When a company's financial performance or compliance issues raise concerns, it may be labeled with the "ST" designation, "*ST" signifies an even higher level of concern regarding the financial health or compliance of a listed company.) companies are excluded; (3) enterprises with missing or abnormal relevant variables. After screening, 31,748 firm-year observations are obtained. In addition, in order to eliminate the influence of outliers, this paper winsorizes all continuous variables at the 1% and 99% levels (Table 1).

The data are obtained from the following sources: (1) Data on employee education level is collected manually from the annual reports of listed companies; (2) Confucian culture data is manually collated through Baidu, Google search engine and China National Studies Network; (3) Other financial data is obtained from the CSMAR(China Stock Market & Accounting Research) database (Table 2).

### Definition of variables

**Audit fees.** Referring to the research of Liu et al., the natural logarithm (LNAF) of audit fees publicly disclose in the annual reports of listed companies is used as the dependent variable in this paper [72].

**Education level of employees.** This paper manually collects the data of the educational level of employees of A-share listed companies, and uses the proportion of the number of employees with bachelor's degree or above to the total number of employees as the measurement variable of the educational level of employees. The reason for the adoption of bachelor's degree or above is that employees with a university education level or below are generally not included in the measurement of human capital according to the literature on human capital measurement. Therefore, employees with a bachelor's degree or above are defined as highly educated employees in this paper.

**Labor protection.** In this paper, the Labor Protection Law implemented in 2008 is selected as the exogenous variable of labor protection intensity. The value of LAW equals 1 for 2008 and subsequent years, and 0 otherwise.

**Control variables.** Referring to the existing literature [73], this paper controls the size of listed company (SIZE), return on assets (ROA), asset-liability ratio (LEV), loss or not (LOSS), book-to-market ratio (BM), inventory and accounts receivable ratio (RIP), quick ratio

**Table 1. Sample selection.**

| | |
|---|---:|
| Initial sample during 2006–2021 | 41,655 |
| Delete: | |
| Financial and insurance listed companies | 1,019 |
| ST and *ST companies are excluded | 2,925 |
| Enterprises with missing or abnormal relevant variables | 5.963 |
| Final sample | 31,748 |

**Table 2. Sample distribution.**

|  | Number | Percent (%) |
|---|---|---|
| Agriculture, forestry, animal husbandry and fishery (A) | 443 | 1.40% |
| Mining industry (B) | 871 | 2.74% |
| Manufacturing industry (C) | 19,400 | 61.11% |
| Electricity, heat, gas and water production and supply (D) | 1,264 | 3.98% |
| Construction industry (E) | 858 | 2.70% |
| Wholesale and retail (F) | 1,890 | 5.95% |
| Transportation, storage and postal industry (G) | 1,058 | 3.33% |
| Accommodation and Catering Industry (H) | 121 | 0.38% |
| Information transmission, software and information technology services (I) | 2,102 | 6.62% |
| Real estate (K) | 1,600 | 5.04% |
| Leasing and business services (L) | 461 | 1.45% |
| Scientific research and technical service industry (M) | 283 | 0.89% |
| Water conservancy, environment and public facilities management industry (N) | 443 | 1.40% |
| Education (P) | 75 | 0.24% |
| Health and social work (Q) | 117 | 0.37% |
| Culture, sports and entertainment industry (R) | 475 | 1.50% |
| Comprehensive (S) | 287 | 0.90% |
| Total | 31,748 | 100.00% |

(QUICK), Big 4 accounting firm (BIG4), and the type of audit opinion in the last period (LAGOPINION), whether the firm accounting has changed (AUDITTURN), nature of property rights (SOE). The definitions of variables are listed in the Table 3.

## Model design

Based on the research of Dao et al., this paper constructs the following regression model (1) to test H1 [74]:

$$\begin{aligned}
LNAF = {} & \beta_0 + \beta_1 EDU + \beta_2 SIZE + \beta_3 ROA + \beta_4 LEV + \beta_5 LOSS + \beta_6 BM + \beta_7 RIP + \beta_8 QUICK \\
& + \beta_9 BIG4 + \beta_{10} LAGOPINION + \beta_{11} AUDITTURN + \beta_{12} SOE + \beta_{13} Gender + \beta_{14} Age \\
& + \beta_{15} Share + \beta_{16} Degree + \beta_{17} Finan + \beta_{18} Tenure + \beta_{19} Change + \beta_{20} StaffDirector \\
& + \gamma Ind + \delta Year + \varepsilon
\end{aligned} \tag{1}$$

In addition, referring to the method of Wang and Zhu (2018), the following regression model (2) is used to test H2:

$$\begin{aligned}
LNAF = {} & \beta_0 + \beta_1 EDU + \beta_2 LAW + \beta_3 EDU \times LAW + \beta_4 SIZE + \beta_5 ROA + \beta_6 LEV + \beta_7 LOSS \\
& + \beta_8 BM + \beta_9 RIP + \beta_{10} QUICK + \beta_{11} BIG4 + \beta_{12} LAGOPINION + \beta_{13} AUDITTURN \\
& + \beta_{14} SOE + \beta_{15} Gender + \beta_{16} Age + \beta_{17} Share + \beta_{18} Degree + \beta_{19} Finan + \beta_{20} Tenure \\
& + \beta_{21} Change + \beta_{22} StaffDirector + \gamma Ind + \delta Year + \varepsilon
\end{aligned} \tag{2}$$

# Empirical results and analysis

## Descriptive statistics

Table 4 reports the descriptive statistics results of each variable. It can be found that the mean, minimum and maximum value of audit fee (LNAF) are 13.6855, 12.0695 and 16.4759. This

**Table 3. Variable definitions.**

| Variable | Variable Definition |
|---|---|
| LNAF | Natural logarithm of audit fees in RMB. |
| EDU | Proportion of the number of employees with bachelor's degree or above to the total number of employees in the enterprise. |
| LAW | The value is 1 for 2008 and subsequent years, and 0 otherwise. |
| SIZE | Natural logarithm of the company's total assets in RMB. |
| ROA | Net profit for divided by total assets. |
| LEV | Liabilities divided by total assets. |
| LOSS | The value is 1 when net profit for the year is negative, and 0 otherwise. |
| BM | Book value of total assets divided by market value. |
| RIP | (Amount of inventory + amount of accounts receivable)/operating income. |
| QUICK | (Current assets—amount of inventory)/current liabilities. |
| BIG4 | The value is 1 when the audit institution is a Big 4 accounting firm, and 0 otherwise. |
| LAGOPINION | The value is 1 if the previous audit opinion is a standard unqualified audit opinion, and 0 otherwise. |
| AUDITTURN | The value is 1 when there is a change in the audit institution within the sample year, and 0 otherwise. |
| SOE | The value is 1 for state-owned enterprises, and 0 otherwise. |
| Gender | The value is 1 if the general manager is male, and 0 otherwise. |
| Age | Age of general manager. |
| Share | The value is 1 if the general manager holds shares of the company at the end of the year, and 0 otherwise. |
| Degree | The value is 1 if the general manager has a bachelor degree or above, and 0 otherwise. |
| Finan | The value is 1 if the general manager has worked in the financial industry, and 0 otherwise. |
| Tenure | Years of auditor working. |
| Change | The value is 1 if the signature auditor changes, and 0 otherwise. |
| StaffDirector | The value is 1 if the company includes staff representatives within its board of directors, and 0 otherwise. |

indicates that the actual values of the audit fees are approximately 808,151.68, 174,166.91, and 13,399,485.33 RMB, respectively, which are close to the study of Zhang et al. [75]. The average and maximum educational level (EDU) of employees are 0.1985 and 0.7522 respectively, indicating that the average number of employees with a bachelor's degree or above accounted for 19.85% of the total number of employees in A-share listed companies in China, and the number of employees with a bachelor's degree or above accounted for 75.22% of the total number of employees in enterprises with the highest educational level. In addition, as shown in S1 Fig, the average ratio of the number of employees with bachelor's degree or above to the total number of employees in A-share listed companies shows an increasing trend year by year due to the development and deepening of the strategy of rejuvenating the country through science and education and strengthening the country through talent.

In terms of control variables, the mean and median value of the size of listed company (SIZE) are 22.0631 and 21.9184. This suggests that the actual values for the company size are approximately 1,093,556,776.7 and 826,429,158.5 RMB, indicating that A-share listed companies exhibit a large size. The mean and median value of return on assets (ROA) are 0.0345 and 0.0353, indicating that the profitability of A-share listed companies is relatively general. The mean and median value of the asset-liability ratio (LEV) are 0.4489 and 0.4435, indicating that A-share listed companies are facing possible financial pressure. The average value of loss or not (LOSS) is 0.1033, indicating that fewer A-share listed companies are facing losses. The mean and median value of book-to-market ratio (BM) are 0.6216 and 0.6249, indicating that

**Table 4. Descriptive statistics.**

| Variable | N | Mean | S.D. | Min | P25 | Median | P75 | Max |
|---|---|---|---|---|---|---|---|---|
| LNAF | 31748 | 13.6855 | 0.7211 | 12.0695 | 13.1910 | 13.5950 | 14.0692 | 16.4759 |
| EDU | 31748 | 0.1985 | 0.1800 | 0.0000 | 0.0601 | 0.1578 | 0.2930 | 0.7522 |
| SIZE | 31748 | 22.0631 | 1.3174 | 18.8533 | 21.1265 | 21.9184 | 22.8302 | 26.4603 |
| ROA | 31748 | 0.0345 | 0.0659 | -0.3022 | 0.0131 | 0.0353 | 0.0650 | 0.2163 |
| LEV | 31748 | 0.4489 | 0.2140 | 0.0489 | 0.2809 | 0.4435 | 0.6049 | 1.0055 |
| LOSS | 31748 | 0.1033 | 0.3044 | 0.0000 | 0.0000 | 0.0000 | 0.0000 | 1.0000 |
| BM | 31748 | 0.6216 | 0.2449 | 0.0443 | 0.4357 | 0.6249 | 0.8115 | 1.1668 |
| RIP | 31748 | 0.6409 | 0.7766 | -0.0737 | 0.2309 | 0.4250 | 0.7340 | 5.3547 |
| QUICK | 31748 | 1.7358 | 2.0625 | 0.1512 | 0.6561 | 1.0860 | 1.8855 | 13.4261 |
| BIG4 | 31748 | 0.0517 | 0.2215 | 0.0000 | 0.0000 | 0.0000 | 0.0000 | 1.0000 |
| LAGOPINION | 31748 | 0.9625 | 0.1900 | 0.0000 | 1.0000 | 1.0000 | 1.0000 | 1.0000 |
| AUDITTURN | 31748 | 0.1279 | 0.3340 | 0.0000 | 0.0000 | 0.0000 | 0.0000 | 1.0000 |
| SOE | 31748 | 0.4117 | 0.4922 | 0.0000 | 0.0000 | 0.0000 | 1.0000 | 1.0000 |
| Gender | 31748 | 0.9389 | 0.2396 | 0.0000 | 1.0000 | 1.0000 | 1.0000 | 1.0000 |
| Age | 31748 | 49.1450 | 6.4755 | 33.0000 | 45.0000 | 49.0000 | 54.0000 | 65.0000 |
| Share | 31748 | 0.4511 | 0.4976 | 0.0000 | 0.0000 | 0.0000 | 1.0000 | 1.0000 |
| Degree | 31748 | 0.6770 | 0.4676 | 0.0000 | 0.0000 | 1.0000 | 1.0000 | 1.0000 |
| Finan | 31748 | 0.0529 | 0.2238 | 0.0000 | 0.0000 | 0.0000 | 0.0000 | 1.0000 |
| Tenure | 31748 | 7.3079 | 5.3106 | 1.0000 | 3.0000 | 6.0000 | 10.0000 | 23.0000 |
| Change | 31748 | 0.6293 | 0.4830 | 0.0000 | 0.0000 | 1.0000 | 1.0000 | 1.0000 |
| StaffDirector | 31748 | 0.0089 | 0.0938 | 0.0000 | 0.0000 | 0.0000 | 0.0000 | 1.0000 |

A-share listed companies are less risky. The mean and median value of inventory and accounts receivable ratio (RIP) are 0.6409 and 0.4250, indicating that A-share listed companies have potential operational risks. The mean and median value of quick ratio (QUICK) are 1.7358 and 1.0860, indicating that A-share listed companies have solvency. The mean value of BIG4 firms (BIG4) is 0.0517, indicating that the market share of the BIG4 firms is about 5.17%. The mean value of the type of audit opinion in the last period (LAGOPINION) is 0.9625, indicating that the vast majority of audit opinions in A-share listed companies are standard unqualified audit opinions. The mean value of whether the firm accounting has changed (AUDITTURN) is 0.1279, indicating that a number of A-share listed companies have changed their accounting firms. The mean value of nature of property rights (SOE) is 0.4117, indicating that 41.17% of the companies in the sample are state-owned enterprises. The mean value of gender of general manager (Gender) is 0.9389, indicating that 93.89% of the companies in the sample is male. The mean and median value of age of general manager (Age) is 49.1450 and 49.0000, indicating that the general managers of A-share companies are generally older. The mean value of general manager holds shares of the company (Share) is 0.4511, indicating that 45.11% of general manager holds shares of the company at the end of the year in the sample. The mean value of the degree of general manager (Degree) is 0.6770, indicating that 67.70% of general manager has a bachelor degree or above in the sample. The mean value of financial background of general manager (Finan) is 0.0529, indicating that 5.29% of general manager has worked in the financial industry in the sample. The mean and median value of the years of auditor working (Tenure) is 7.3079 and 6.0000. The mean value of auditor change (Change) is 0.6293, indicating that 62.93% of signature auditor changes in the sample. The mean value of staff representative directors (StaffDirectoris) is 0.0089, indicating that 0.89% of companies have staff representative directors. In general, the statistical characteristics of the main control variables are basically consistent with the existing literature.

## Staff education level and audit fees

Table 5 reports the regression results of employee education level and audit fees. If employee education level (EDU) is a negative value, meaning employees with high educational background can effectively reduce the audit fees borne by the company. Of which, column (1) shows the regression results using the OLS model, and column (2) shows the regression results using the fixed effects model. It can be found from the results in the table that the coefficients of employee education level (EDU) and audit fee (LNAF) are significantly negative at the level of 1% or 5% in columns (1) and (2), which means that employee education level has a significant inhibitory effect on audit fee. Evidence supports Hypothesis 1.

In terms of control variables, the size of listed company (SIZE) and Big 4 accounting firm (BIG4) are significantly positively correlated with audit fees, indicating that the audit fees of companies with large size and Big 4 accounting firm are higher. Return on assets (ROA), book-to-market ratio (BM), inventory and accounts receivable ratio (RIP), quick ratio (QUICK), the type of audit opinion in the last period (LAGOPINION), whether the firm accounting has changed (AUDITTURN), nature of property rights (SOE), and the years of auditor working (Tenure) are significantly negatively correlated with audit fees. It indicates that the audit fees of the companies with better profitability, higher book-to-market ratio, lower operating risk, stronger solvency, lower audit risk in the previous year, changed accounting firms are lower, and more years of auditor working.

## Employee education level, labor protection intensity and audit fees

Table 6 reports the regression results of employee education level, labor protection intensity and audit fees. If the summation of EDU + LAW + EDU × LAW yields a positive value, indicating that the enactment of the Law surpasses the effect of EDU. Where, column (1) shows the regression results using OLS model, and column (2) shows the regression results using fixed effects model. According to the results in the table, the coefficient of EDU×LAW is significantly positive at the 1% level in columns (1) and (2). This indicates that the implementation of the Labor Protection Law has significantly weakened the inhibitory effect of highly educated employees on audit fees. Evidence supports Hypothesis 2.

## Robustness test

### Endogeneity problem

The model (1) in this paper may have the following endogeneity problems: (1) The problem of reverse causality. That is, enterprises hire more highly educated employees in order to reduce audit fees, so the negative correlation of model (1) does not show that highly educated employees reduce audit fees. (2) The problem of omitted variables, which is a common problem in empirical studies, may lead to obvious bias in the regression results of main explanatory variables. In view of the endogeneity problem, this paper refers to the existing research and uses the GDP per capita (GDP_P) and unemployment rate (UNEMPLOYMENT) in the t year of the province where the listed company is located as instrumental variables, and adopts the two-stage least squares method to carry out the test. The corresponding test results are reported in Tables 7 and 8. The reasons for choosing annual GDP per capita (GDP_P) and unemployment rate (UNEMPLOYMENT) in the province where the listed company resides as instrumental variables are as follows: (1) The higher the GDP per capita (GDP_P) and the lower the unemployment rate (UNEMPLOYMENT), the more highly educated people will be attracted to work in the region. Therefore, the GDP per capita (GDP_P) and unemployment rate (UNEMPLOYMENT) both significantly affect the education level (EDU) of the employees

**Table 5. Employee education level and audit fees.**

| Variable | LNAF | |
|---|---|---|
| | **OLS Model** | **Fixed effects Model** |
| | **(1)** | **(2)** |
| EDU | -0.1106*** | -0.0654** |
| | (-3.07) | (-2.19) |
| SIZE | 0.3777*** | 0.3015*** |
| | (47.10) | (42.64) |
| ROA | -0.4745*** | -0.1873*** |
| | (-5.18) | (-2.98) |
| LEV | 0.0146 | 0.0851** |
| | (0.37) | (2.55) |
| LOSS | 0.0292** | 0.0198** |
| | (1.97) | (2.14) |
| BM | -0.1914*** | 0.0201 |
| | (-6.23) | (0.98) |
| RIP | -0.0381*** | -0.0260*** |
| | (-4.85) | (-4.08) |
| QUICK | -0.0132*** | -0.0077*** |
| | (-4.65) | (-3.62) |
| BIG4 | 0.6760*** | 0.2678*** |
| | (17.37) | (8.06) |
| LAGOPINION | -0.1885*** | -0.1048*** |
| | (-8.11) | (-5.95) |
| AUDITTURN | -0.0432*** | -0.0245*** |
| | (-4.76) | (-4.11) |
| SOE | -0.0705*** | -0.0243 |
| | (-4.75) | (-1.47) |
| Gender | 0.0277 | -0.0154 |
| | (1.43) | (-1.20) |
| Age | 0.0010 | 0.0008* |
| | (1.35) | (1.73) |
| Share | -0.0172 | -0.0093 |
| | (-1.60) | (-1.25) |
| Degree | -0.0082 | 0.0091 |
| | (-0.75) | (1.29) |
| Finan | 0.0152 | 0.0063 |
| | (0.87) | (0.59) |
| Tenure | 0.0037*** | -0.0010 |
| | (3.48) | (-1.22) |
| Change | -0.0064 | -0.0032 |
| | (-1.37) | (-1.07) |
| StaffDirector | -0.0011 | -0.0409 |
| | (-0.02) | (-1.50) |
| Constant | 5.4397*** | 6.7484*** |
| | (33.94) | (44.57) |
| IND | YES | YES |
| YEAR | YES | YES |
| N | 31748 | 31748 |

(*Continued*)

**Table 5.** (Continued)

| Variable | LNAF | |
|---|---|---|
| | **OLS Model** | **Fixed effects Model** |
| | **(1)** | **(2)** |
| *adj.R²* | 0.6667 | 0.6668 |

Note (1)

***, ** and * indicate significant at 1%, 5% and 10% levels, respectively; (2) Values in parentheses are t-values adjusted for heteroscedasticity.

of the listed company. The indicators meet the requirement that instrumental variables should be correlated; (2) The GDP per capita (GDP_P) and unemployment rate (UNEMPLOY-MENT) in the t year of each province are not affected by the audit fees of enterprises, and the index meets the exogeneity requirement of the instrumental variables.

In the 2SLS test for Hypothesis 1, this paper uses the control variables mentioned above and the instrumental variables mentioned above to regression the education level (EDU) of employees in the first stage. The results in column (1) of Table 7 show that GDP per capita (GDP_P) is significantly positively correlated with the education level of employees at the level of 1%, while unemployment rate (UNEMPLOYMENT) is significantly negatively correlated with the education level of employees at the level of 1%, indicating that the higher the GDP per capita (GDP_P) and the lower the unemployment rate (UNEMPLOYMENT), the more highly educated people will be attracted to work in the region. Then, in the second stage, the predicted value of EDU is used as an explanatory variable to test the model (1). The results in columns (2) and (3) of Table 7 show that the predicted value of the education level (EDU) is significantly negatively correlated with audit fees at the 1% level. The above results indicate that the conclusion of the H1 study is robust after controlling for endogeneity.

Further, in the 2SLS test for hypothesis 2, this paper first uses the control variables mentioned above and the instrumental variables mentioned above to regression the education level (EDU) of employees in the first stage. The results in column (1) of Table 8 show that GDP per capita (GDP_P) is significantly positively correlated with the education level of employees at the level of 1%, while unemployment rate (UNEMPLOYMENT) is significantly negatively correlated with the education level (EDU) of employees at the level of 1%, indicating that the higher the GDP per capita (GDP_P) and the lower the unemployment rate (UNEMPLOY-MENT), the more highly educated people will be attracted to work in the region. Then, in the second stage, the predicted value of the education level (EDU) is used as an explanatory variable to test the model (2). The results in columns (2) and (3) of Table 8 show that the coefficients of EDU×LAW are all significantly positive at the 1% level. The above results indicate that the conclusion of H2 study is assumed to be robust after controlling for endogeneity.

## Sensitivity test of measurement index

In the previous study, the proportion of the number of employees with bachelor's degree or above to the total number of employees in the enterprise is used to measure the education level of employees. Here, this paper measures the education level of employees by the proportion of the number of employees with a graduate degree or above to the total number of employees in the enterprise, and conducts a sensitivity test on the results of hypothesis 1. The results in Table 9 show that the coefficient of employee education level (MAS) and audit expense (LNAF) is significantly negative at the 1% level in columns (1) and (2). The above results

**Table 6. Employee education level, labor protection intensity and audit fees.**

| Variable | LNAF | |
|---|---|---|
| | OLS Model | Fixed effects Model |
| | (1) | (2) |
| EDU | -0.4292*** | -0.2196*** |
| | (-6.45) | (-4.51) |
| LAW | 0.3808*** | 0.5460*** |
| | (17.93) | (29.80) |
| EDU×LAW | 0.3400*** | 0.1731*** |
| | (5.26) | (3.57) |
| SIZE | 0.3771*** | 0.3009*** |
| | (46.96) | (42.54) |
| ROA | -0.4705*** | -0.1855*** |
| | (-5.15) | (-2.96) |
| LEV | 0.0144 | 0.0849** |
| | (0.36) | (2.55) |
| LOSS | 0.0297** | 0.0201** |
| | (2.00) | (2.18) |
| BM | -0.1898*** | 0.0207 |
| | (-6.18) | (1.01) |
| RIP | -0.0383*** | -0.0260*** |
| | (-4.88) | (-4.09) |
| QUICK | -0.0133*** | -0.0077*** |
| | (-4.70) | (-3.64) |
| BIG4 | 0.6764*** | 0.2695*** |
| | (17.37) | (8.12) |
| LAGOPINION | -0.1885*** | -0.1051*** |
| | (-8.11) | (-5.97) |
| AUDITTURN | -0.0434*** | -0.0246*** |
| | (-4.77) | (-4.13) |
| SOE | -0.0705*** | -0.0246 |
| | (-4.75) | (-1.49) |
| Gender | 0.0272 | -0.0155 |
| | (1.41) | (-1.21) |
| Age | 0.0009 | 0.0008* |
| | (1.32) | (1.72) |
| Share | -0.0175 | -0.0095 |
| | (-1.64) | (-1.28) |
| Degree | -0.0084 | 0.0089 |
| | (-0.76) | (1.28) |
| Finan | 0.0158 | 0.0067 |
| | (0.90) | (0.62) |
| Tenure | 0.0037*** | -0.0009 |
| | (3.49) | (-1.19) |
| Change | -0.0064 | -0.0032 |
| | (-1.38) | (-1.07) |
| StaffDirector | -0.0019 | -0.0413 |
| | (-0.04) | (-1.51) |
| Constant | 5.4853*** | 6.7759*** |

*(Continued)*

**Table 6.** (Continued)

| Variable | LNAF | |
|---|---|---|
| | **OLS Model** | **Fixed effects Model** |
| | **(1)** | **(2)** |
| | (34.12) | (44.70) |
| IND | YES | YES |
| YEAR | YES | YES |
| N | 31748 | 31748 |
| adj.R² | 0.6670 | 0.6670 |

Note (1)

\*\*\*, \*\* and \* indicate significant at 1%, 5% and 10% levels, respectively; (2) Values in parentheses are t-values adjusted for heteroscedasticity.

indicate that the conclusion of hypothesis 1 study remains robust in the sensitivity test of measurement indicators. In the aforementioned research, this paper conducts clustering processing at the firm and year level. Here, this paper also carries out clustering processing at the industry level to test the sensitivity of the estimation method of hypothesis 1 research results. The results show that the coefficient of employee education level (EDU) and audit fee (LNAF) is significantly negative at the 1% or 10% level. These results indicate that the conclusion of the hypothesis 1 remains robust to the sensitivity test of the estimation method.

Furthermore, the proportion of graduate degree or above in the total number of employees in the enterprise is used to measure the education level of employees, and the sensitivity test of the measurement index of hypothesis 2 research results is conducted. The results in Table 10 show that the coefficient of EDU×LAW is significantly positive at the 1% level in columns (1) and (2). The above results indicate that the conclusion of the hypothesis 2 study remains robust in the sensitivity test of the metric indicators. Further, this paper also carries out industry-level clustering processing to test the sensitivity of the estimation method of hypothesis 2 research results. The results show that the coefficient of EDU×LAW is significantly positive at the 1% level. These results indicate that the conclusion of the hypothesis 2 is robust to the sensitivity test of the estimation method.

## Mediation effect test

If the employee education level can affect audit fees, it will inevitably trigger more profound thinking: through what transmission mechanism does employee education level affect audit fees? Employees with higher education level generally improve the quality of financial reporting by learning and using new technologies [57] and improving aware of supervision [58]. On the other hand, superior reporting quality lowers audit risk and the need for greater audit investments, such as auditors charge family firms less [63]. Therefore, this paper speculates that the quality of financial reporting can play an intermediary effect between employee education level and audit fees of listed companies.

Mediator Effect is an important statistical concept, which means that the explanatory variable (X) and the explained variable (Y) are not directly causal but indirectly affected through variable (M). At this time, variable (M) is called a mediator variable, and X becomes a mediator effect through M's indirect influence on Y. The corresponding relations are as follows: $Y = c \times X + e1$、$M = a \times X + e2$、$Y = c' \times X + b \times M + e3$. This paper draws on the methods of Judd and Kenny, Baron and Kenny and Wen Zhonglin et al. to carry out mediation effect analysis through three steps: Path a, Path b and Path c [76–78].

**Table 7. 2SLS Test (H1).**

| Variable | First Stage | Second Stage | |
|---|---|---|---|
| | EDU | LNAF | |
| | (1) | (2) | (3) |
| GDP_P | 0.0066*** | | |
| | (3.15) | | |
| UNEMPLOYMENT | -0.0173*** | | |
| | (-5.87) | | |
| EDU_IV | | -2.1226*** | -1.0749*** |
| | | (-7.87) | (-4.76) |
| SIZE | 0.0171*** | 0.4009*** | 0.3153*** |
| | (7.52) | (47.56) | (41.93) |
| ROA | 0.0407 | -0.3600*** | -0.1459** |
| | (1.19) | (-3.93) | (-2.28) |
| LEV | 0.0244* | 0.0508 | 0.1076*** |
| | (1.80) | (1.26) | (3.18) |
| LOSS | -0.0153*** | 0.0030 | 0.0049 |
| | (-2.99) | (0.19) | (0.51) |
| BM | -0.0950*** | -0.3663*** | -0.0750** |
| | (-9.42) | (-9.42) | (-2.54) |
| RIP | 0.0345*** | 0.0298** | 0.0085 |
| | (10.97) | (2.45) | (0.85) |
| QUICK | 0.0117*** | 0.0086** | 0.0038 |
| | (9.53) | (2.07) | (1.15) |
| BIG4 | 0.0172 | 0.6458*** | 0.2598*** |
| | (1.60) | (17.63) | (7.95) |
| LAGOPINION | 0.0062 | -0.1720*** | -0.0977*** |
| | (0.77) | (-7.25) | (-5.54) |
| AUDITTURN | -0.0010 | -0.0436*** | -0.0249*** |
| | (-0.34) | (-4.84) | (-4.22) |
| SOE | 0.0255*** | -0.0252 | -0.0007 |
| | (4.99) | (-1.61) | (-0.04) |
| Gender | 0.0032 | 0.0320* | -0.0136 |
| | (0.46) | (1.67) | (-1.07) |
| Age | -0.0003 | 0.0002 | 0.0004 |
| | (-1.28) | (0.22) | (0.89) |
| Share | 0.0113*** | 0.0049 | 0.0010 |
| | (3.05) | (0.44) | (0.13) |
| Degree | 0.0187*** | 0.0288** | 0.0278*** |
| | (5.02) | (2.44) | (3.41) |
| Finan | -0.0185*** | -0.0189 | -0.0110 |
| | (-2.63) | (-1.06) | (-0.97) |
| Tenure | -0.0017*** | 0.0004 | -0.0026*** |
| | (-4.42) | (0.31) | (-2.89) |
| Change | -0.0001 | -0.0069 | -0.0036 |
| | (-0.08) | (-1.49) | (-1.21) |
| StaffDirector | 0.0285 | 0.0428 | -0.0196 |
| | (1.39) | (0.93) | (-0.69) |
| Constant | -0.3160*** | 5.0555*** | 6.5129*** |

(*Continued*)

**Table 7.** (Continued)

| Variable | First Stage | Second Stage | |
|---|---|---|---|
| | EDU | LNAF | |
| | (1) | (2) | (3) |
| | (-6.19) | (30.77) | (41.45) |
| IND | YES | YES | YES |
| YEAR | YES | YES | YES |
| N | 31748 | 31748 | 31748 |
| adj.R$^2$ | 0.3412 | 0.6720 | 0.6679 |
| F Value-IV | 19.2249 | | |

Note (1)

***, ** and * indicate significant at 1%, 5% and 10% levels, respectively; (2) Values in parentheses are t-values adjusted for heteroscedasticity.

The specific testing steps are as follows: (1) Path a is used to test the relationship between employee education level and audit fees. If $\alpha 1$ is statistically significant, the next step is carried out. (2) Path b was used to test the relationship between employee education level and quality of financial reporting. We selected DA, IC and Restatement respectively to measure the quality of financial reporting. If $\beta 1$ was statistically significant, the next step was carried out. (3) In this paper, Path c is used to analyze the mediation effect, and employee education level is included into the model. If $\lambda 1$ is small relative to $\alpha 1$ in absolute value, and $\lambda 1$ is no longer significant, the complete mediation effect is proved to be valid; if $\lambda 1$ is still significant and only decreases in absolute value, the partial mediation effect is proved to be valid.

This paper tests whether employee education level can affect the quality of financial reporting, and then affect audit fees. The explained variable (LNAF) is the natural logarithm of audit fees. The explanatory variable (EDU) is the proportion of the number of employees with bachelor's degree or above to the total number of employees in the enterprise. The definitions of mediation variables are listed in the Table 11.

Path a: $LNAF_{i,t} = \alpha_0 + \alpha_1 EDU_{i,t} + \alpha_2 Controls_{i,t} + IND + \varepsilon_{i,t}$ (3)

Path b: $DA_{i,t} = \beta_0 + \beta_1 EDU_{i,t} + \beta_2 Controls_{i,t} + IND + \varepsilon_{i,t}$ (4)

Path c: $LNAF_{i,t} = \lambda_0 + \lambda_1 EDU_{i,t} + \lambda_2 DA_{i,t} + \lambda_3 Controls_{i,t} + IND + \varepsilon_{i,t}$ (5)

In the above equation, Table 12 lists the regression results about discretionary accruals (DA). Path a results are as described above and will not be repeated here. In Path b, employee education level (EDU) is significantly negatively correlated with discretionary accruals (DA) at the level of 1% in OLS Model and Fixed effects Model, indicating that the higher the employee education level (EDU), the less discretionary accruals (DA). In Path c, the coefficients of employee education level (EDU) are -0.1085 at the level of 1% in OLS Model and -0.0656 at the level of 5% in Fixed effects Model respectively, but the absolute value are smaller than the coefficient in Path a. The results of Sobel Z test are all significant, indicating that part of the mediation effect was established. The above results indicate that employee education level can reduce audit fees by improving the quality of financial reporting (reducing discretionary accruals).

Path a: $LNAF_{i,t} = \alpha_0 + \alpha_1 EDU_{i,t} + \alpha_2 Controls_{i,t} + IND + \varepsilon_{i,t}$ (3)

Path b: $IC_{i,t} = \beta_0 + \beta_1 EDU_{i,t} + \beta_2 Controls_{i,t} + IND + \varepsilon_{i,t}$ (4)

Path c: $LNAF_{i,t} = \lambda_0 + \lambda_1 EDU_{i,t} + \lambda_2 IC_{i,t} + \lambda_3 Controls_{i,t} + IND + \varepsilon_{i,t}$ (5)

**Table 8. 2SLS test (H2).**

| Variable | First Stage | Second Stage | |
|---|---|---|---|
| | EDU | LNAF | |
| | (1) | (2) | (3) |
| GDP_P | 0.0066*** | | |
| | (3.15) | | |
| UNEMPLOYMENT | -0.0173*** | | |
| | (-5.87) | | |
| EDU_IV | | -1.9101*** | -1.1133*** |
| | | (-7.07) | (-4.85) |
| LAW | | 0.8295*** | 0.7784*** |
| | | (23.37) | (24.69) |
| EDU_IV*LAW | | 0.8871*** | 0.4235*** |
| | | (5.93) | (3.48) |
| SIZE | 0.0171*** | 0.4014*** | 0.3193*** |
| | (7.52) | (50.96) | (44.63) |
| ROA | 0.0407 | -0.3143*** | -0.1305** |
| | (1.19) | (-3.51) | (-2.08) |
| LEV | 0.0244* | 0.0674* | 0.1178*** |
| | (1.80) | (1.71) | (3.51) |
| LOSS | -0.0153*** | -0.0057 | -0.0017 |
| | (-2.99) | (-0.38) | (-0.18) |
| BM | -0.0950*** | -0.4107*** | -0.1105*** |
| | (-9.42) | (-11.51) | (-4.07) |
| RIP | 0.0345*** | 0.0503*** | 0.0221** |
| | (10.97) | (4.68) | (2.45) |
| QUICK | 0.0117*** | 0.0153*** | 0.0085*** |
| | (9.53) | (4.12) | (2.77) |
| BIG4 | 0.0172 | 0.6089*** | 0.2524*** |
| | (1.60) | (17.47) | (7.91) |
| LAGOPINION | 0.0062 | -0.1611*** | -0.0928*** |
| | (0.77) | (-6.80) | (-5.30) |
| AUDITTURN | -0.0010 | -0.0423*** | -0.0249*** |
| | (-0.34) | (-4.76) | (-4.25) |
| SOE | 0.0255*** | -0.0115 | 0.0097 |
| | (4.99) | (-0.76) | (0.59) |
| Gender | 0.0032 | 0.0327* | -0.0126 |
| | (0.46) | (1.73) | (-1.00) |
| Age | -0.0003 | -0.0001 | 0.0003 |
| | (-1.28) | (-0.11) | (0.60) |
| Share | 0.0113*** | 0.0110 | 0.0055 |
| | (3.05) | (1.03) | (0.74) |
| Degree | 0.0187*** | 0.0395*** | 0.0353*** |
| | (5.02) | (3.45) | (4.50) |
| Finan | -0.0185*** | -0.0298* | -0.0180 |
| | (-2.63) | (-1.72) | (-1.60) |
| Tenure | -0.0017*** | -0.0007 | -0.0033*** |
| | (-4.42) | (-0.60) | (-3.84) |
| Change | -0.0001 | -0.0070 | -0.0039 |

*(Continued)*

**Table 8.** (Continued)

| Variable | First Stage | Second Stage | |
|---|---|---|---|
| | EDU | LNAF | |
| | (1) | (2) | (3) |
| | (-0.08) | (-1.52) | (-1.30) |
| StaffDirector | 0.0285 | 0.0562 | -0.0087 |
| | (1.39) | (1.26) | (-0.31) |
| Constant | -0.3160*** | 4.9874*** | 6.4115*** |
| | (-6.19) | (32.12) | (42.48) |
| IND | YES | YES | YES |
| YEAR | YES | YES | YES |
| N | 31748 | 31748 | 31748 |
| adj.R$^2$ | 0.3412 | 0.6792 | 0.6704 |
| F Value-IV | 10.1118 | | |

Note (1)

***, ** and * indicate significant at the 1%, 5% and 10% levels, respectively; (2) Values in parentheses are t-values adjusted for heteroscedasticity.

**Table 9. Sensitivity test of measurement indicators (H1).**

| Variable | LNAF | |
|---|---|---|
| | OLS Model | Fixed effects Model |
| | (1) | (2) |
| MAS | -0.4960*** | -0.2863** |
| | (-3.99) | (-2.54) |
| SIZE | 0.3783*** | 0.3015*** |
| | (46.97) | (42.96) |
| ROA | -0.4804*** | -0.1892*** |
| | (-5.27) | (-3.02) |
| LEV | 0.0107 | 0.0849** |
| | (0.27) | (2.55) |
| LOSS | 0.0289* | 0.0198** |
| | (1.95) | (2.15) |
| BM | -0.2003*** | 0.0185 |
| | (-6.47) | (0.90) |
| RIP | -0.0371*** | -0.0259*** |
| | (-4.74) | (-4.03) |
| QUICK | -0.0132*** | -0.0076*** |
| | (-4.64) | (-3.60) |
| BIG4 | 0.6786*** | 0.2703*** |
| | (17.47) | (8.13) |
| LAGOPINION | -0.1890*** | -0.1057*** |
| | (-8.19) | (-6.02) |
| AUDITTURN | -0.0426*** | -0.0245*** |
| | (-4.68) | (-4.10) |
| SOE | -0.0670*** | -0.0231 |
| | (-4.49) | (-1.39) |

*(Continued)*

**Table 9.** (Continued)

| Variable | LNAF | |
|---|---|---|
| | **OLS Model** | **Fixed effects Model** |
| | **(1)** | **(2)** |
| Gender | 0.0286 | -0.0151 |
| | (1.48) | (-1.18) |
| Age | 0.0010 | 0.0008* |
| | (1.42) | (1.75) |
| Share | -0.0171 | -0.0097 |
| | (-1.60) | (-1.30) |
| Degree | -0.0083 | 0.0091 |
| | (-0.75) | (1.29) |
| Finan | 0.0174 | 0.0068 |
| | (1.00) | (0.64) |
| Tenure | 0.0037*** | -0.0010 |
| | (3.50) | (-1.24) |
| Change | -0.0063 | -0.0031 |
| | (-1.34) | (-1.04) |
| StaffDirector | -0.0067 | -0.0434 |
| | (-0.14) | (-1.61) |
| Constant | 5.4460*** | 6.7565*** |
| | (33.92) | (44.84) |
| IND | YES | YES |
| YEAR | YES | YES |
| N | 31748 | 31748 |
| adj.R² | 0.6671 | 0.6668 |

Note (1)

***, ** and * indicate significant at 1%, 5% and 10% levels, respectively; (2) Values in parentheses are t-values adjusted for heteroscedasticity.

In the above equation, Table 13 lists the regression results about defects in the internal control (IC). Path a results are as described above and will not be repeated here. In Path b, employee education level (EDU) is significantly negatively correlated with defects in the internal control (IC) at the level of 1% in OLS Model and Fixed effects Model, indicating that the higher the employee education level (EDU), the less defects in the internal control (IC). In Path c, the coefficients of employee education level (EDU) are -0.1036 at the level of 1% in OLS Model and -0.0648 at the level of 5% in Fixed effects Model respectively, but the absolute value are smaller than the coefficient in Path a. The results of Sobel Z test are all significant, indicating that part of the mediation effect was established. The above results indicate that employee education level can reduce audit fees by improving the quality of financial reporting (reducing defects in the internal control).

Path a: $LNAF_{i,t} = \alpha_0 + \alpha_1 EDU_{i,t} + \alpha_2 Controls_{i,t} + IND + \varepsilon_{i,t}$ (3)

Path b: $Restatement_{i,t} = \beta_0 + \beta_1 EDU_{i,t} + \beta_2 Controls_{i,t} + IND + \varepsilon_{i,t}$ (4)

Path c: $LNAF_{i,t} = \lambda_0 + \lambda_1 EDU_{i,t} + \lambda_2 Restatement_{i,t} + \lambda_3 Controls_{i,t} + IND + \varepsilon_{i,t}$ (5)

In the above equation, Table 14 lists the regression results about defects financial restatement (Restatement). Path a results are as described above and will not be repeated here. In Path b, employee education level (EDU) is significantly negatively correlated with financial

**Table 10. Sensitivity test of measurement indicators (H2).**

| Variable | LNAF | |
| --- | --- | --- |
| | OLS Model | Fixed effects Model |
| | (1) | (2) |
| MAS | -2.6060*** | -1.4338*** |
| | (-12.80) | (-9.01) |
| LAW | 0.2922*** | 0.4935*** |
| | (13.52) | (25.73) |
| MAS×LAW | 2.2987*** | 1.3718*** |
| | (10.15) | (7.51) |
| SIZE | 0.3757*** | 0.2996*** |
| | (46.55) | (42.81) |
| ROA | -0.4748*** | -0.1882*** |
| | (-5.23) | (-3.02) |
| LEV | 0.0103 | 0.0845** |
| | (0.26) | (2.55) |
| LOSS | 0.0296** | 0.0203** |
| | (2.00) | (2.21) |
| BM | -0.1925*** | 0.0227 |
| | (-6.22) | (1.11) |
| RIP | -0.0375*** | -0.0259*** |
| | (-4.82) | (-4.04) |
| QUICK | -0.0137*** | -0.0079*** |
| | (-4.84) | (-3.73) |
| BIG4 | 0.6804*** | 0.2759*** |
| | (17.50) | (8.37) |
| LAGOPINION | -0.1888*** | -0.1057*** |
| | (-8.23) | (-6.04) |
| AUDITTURN | -0.0431*** | -0.0248*** |
| | (-4.74) | (-4.15) |
| SOE | -0.0683*** | -0.0244 |
| | (-4.59) | (-1.48) |
| Gender | 0.0278 | -0.0150 |
| | (1.45) | (-1.17) |
| Age | 0.0010 | 0.0008* |
| | (1.36) | (1.72) |
| Share | -0.0183* | -0.0104 |
| | (-1.71) | (-1.39) |
| Degree | -0.0086 | 0.0090 |
| | (-0.78) | (1.29) |
| Finan | 0.0180 | 0.0070 |
| | (1.04) | (0.65) |
| Tenure | 0.0038*** | -0.0009 |
| | (3.54) | (-1.12) |
| Change | -0.0064 | -0.0032 |
| | (-1.38) | (-1.08) |
| StaffDirector | -0.0060 | -0.0432 |
| | (-0.13) | (-1.59) |
| Constant | 5.5986*** | 6.8510*** |

(*Continued*)

**Table 10.** (Continued)

| Variable | LNAF | |
|---|---|---|
| | **OLS Model** | **Fixed effects Model** |
| | **(1)** | **(2)** |
| | (34.52) | (45.52) |
| IND | YES | YES |
| YEAR | YES | YES |
| N | 31748 | 31748 |
| adj.R² | 0.6685 | 0.6681 |

Note (1)

\*\*\*, \*\* and \* indicate significant at 1%, 5% and 10% levels, respectively; (2) Values in parentheses are t-values adjusted for heteroscedasticity.

restatement (*Restatement*) at the level of 1% in OLS Model and Fixed effects Model, indicating that the higher the employee education level (EDU), the less financial restatement (*Restatement*). In Path c, the coefficients of employee education level (EDU) are -0.1094 at the level of 1% in OLS Model and -0.0657 at the level of 5% in Fixed effects Model respectively, but the absolute value are smaller than the coefficient in Path a. The results of Sobel Z test are all significant, indicating that part of the mediation effect was established. The above results indicate that employee education level can reduce audit fees by improving the quality of financial reporting (reducing financial restatement).

## Further tests

### Marketization level, staff education level and audit fees

According to the above analysis, the education level of employees is helpful to reduce audit risks, and then affects audit fees. However, if the audit risk is well controlled by other influencing factors, the role of employee education level will be relatively small. On the contrary, the education level of employees may play a greater role. In this regard, this paper analyzes the impact of the marketization process. Up to now, China's marketization process has achieved universally recognized success, but the process is very uneven. Due to the differences in resource endowment, geographical location and national policies, the degree of marketization varies greatly among regions. In some provinces, especially coastal ones, decisive progress has been made in marketization, while in others, non-market factors still play a very important role in the economy. As the level of marketization changes, the audit risks faced by auditors will vary [79]. Specifically, in regions with higher marketization process, investors have more thorough legal protection, more thorough legal punishment mechanism, and more stringent regulatory environment, which will encourage enterprises to improve the quality of financial reports, reduce the risk of material misstatement in financial reports, and thus reduce audit risks [80]. In addition, the improvement of marketization process means the reduction or even

**Table 11. Mediation variable definitions.**

| Variable | Variable Definition |
|---|---|
| DA | Discretionary accruals calculated by Modified Jones model. |
| IC | The value is 1 if there are defects in the internal control of the listed company, and 0 otherwise. |
| Restatement | The value is 1 if the listed company in subsequent years to the current financial restatement, and 0 otherwise. |

**Table 12. Mediation testing based on DA.**

| Variable | Path a | Path b | Path c | Path a | Path b | Path c |
|---|---|---|---|---|---|---|
| | LNAF | DA | LNAF | LNAF | DA | LNAF |
| | OLS Model | | | Fixed effects Model | | |
| | (1) | (2) | (3) | (4) | (5) | (6) |
| EDU | -0.1106*** | -0.0215*** | -0.1085*** | -0.0654** | -0.0208*** | -0.0648** |
| | (-3.07) | (-4.60) | (-3.02) | (-2.19) | (-4.44) | (-2.17) |
| DA | | | 0.0956*** | | | 0.0491*** |
| | | | (3.58) | | | (2.88) |
| SIZE | 0.3777*** | 0.0056*** | 0.3772*** | 0.3015*** | 0.0055*** | 0.3013*** |
| | (47.10) | (6.18) | (46.98) | (42.64) | (6.10) | (42.58) |
| ROA | -0.4745*** | 0.4706*** | -0.5194*** | -0.1873*** | 0.4755*** | -0.2128*** |
| | (-5.18) | (26.76) | (-5.56) | (-2.98) | (27.05) | (-3.37) |
| LEV | 0.0146 | -0.0215*** | 0.0166 | 0.0851** | -0.0221*** | 0.0869*** |
| | (0.37) | (-3.86) | (0.42) | (2.55) | (-3.94) | (2.60) |
| LOSS | 0.0292** | -0.0045 | 0.0297** | 0.0198** | -0.0043 | 0.0198** |
| | (1.97) | (-1.56) | (1.99) | (2.14) | (-1.47) | (2.15) |
| BM | -0.1914*** | 0.0147*** | -0.1929*** | 0.0201 | 0.0157*** | 0.0191 |
| | (-6.23) | (3.60) | (-6.28) | (0.98) | (3.88) | (0.93) |
| RIP | -0.0381*** | 0.0199*** | -0.0400*** | -0.0260*** | 0.0199*** | -0.0269*** |
| | (-4.85) | (13.25) | (-5.08) | (-4.08) | (13.18) | (-4.21) |
| QUICK | -0.0132*** | 0.0000 | -0.0132*** | -0.0077*** | 0.0000 | -0.0077*** |
| | (-4.65) | (0.08) | (-4.65) | (-3.62) | (0.08) | (-3.63) |
| BIG4 | 0.6760*** | -0.0138*** | 0.6774*** | 0.2678*** | -0.0139*** | 0.2685*** |
| | (17.37) | (-4.15) | (17.39) | (8.06) | (-4.21) | (8.09) |
| LAGOPINION | -0.1885*** | 0.0016 | -0.1886*** | -0.1048*** | 0.0014 | -0.1048*** |
| | (-8.11) | (0.31) | (-8.12) | (-5.95) | (0.28) | (-5.96) |
| AUDITTURN | -0.0432*** | 0.0004 | -0.0433*** | -0.0245*** | 0.0003 | -0.0245*** |
| | (-4.76) | (0.19) | (-4.76) | (-4.11) | (0.15) | (-4.11) |
| SOE | -0.0705*** | -0.0005 | -0.0705*** | -0.0243 | -0.0006 | -0.0242 |
| | (-4.75) | (-0.31) | (-4.75) | (-1.47) | (-0.32) | (-1.47) |
| Gender | 0.0277 | 0.0022 | 0.0275 | -0.0154 | 0.0019 | -0.0154 |
| | (1.43) | (0.72) | (1.42) | (-1.20) | (0.63) | (-1.20) |
| Age | 0.0010 | -0.0002 | 0.0010 | 0.0008* | -0.0002 | 0.0008* |
| | (1.35) | (-1.51) | (1.38) | (1.73) | (-1.46) | (1.75) |
| Share | -0.0172 | -0.0003 | -0.0171 | -0.0093 | -0.0004 | -0.0093 |
| | (-1.60) | (-0.21) | (-1.60) | (-1.25) | (-0.30) | (-1.25) |
| Degree | -0.0082 | 0.0022 | -0.0085 | 0.0091 | 0.0022 | 0.0090 |
| | (-0.75) | (1.54) | (-0.77) | (1.29) | (1.50) | (1.28) |
| Finan | 0.0152 | -0.0009 | 0.0153 | 0.0063 | -0.0012 | 0.0065 |
| | (0.87) | (-0.30) | (0.88) | (0.59) | (-0.40) | (0.60) |
| Tenure | 0.0037*** | -0.0002* | 0.0037*** | -0.0010 | -0.0003** | -0.0009 |
| | (3.48) | (-1.95) | (3.51) | (-1.22) | (-2.02) | (-1.20) |
| Change | -0.0064 | 0.0004 | -0.0065 | -0.0032 | 0.0003 | -0.0032 |
| | (-1.37) | (0.34) | (-1.38) | (-1.07) | (0.26) | (-1.06) |
| StaffDirector | -0.0011 | 0.0013 | -0.0012 | -0.0409 | 0.0011 | -0.0408 |
| | (-0.02) | (0.18) | (-0.03) | (-1.50) | (0.16) | (-1.49) |
| Constant | 5.4397*** | -0.1463*** | 5.4537*** | 6.7484*** | -0.1447*** | 6.7540*** |
| | (33.94) | (-7.68) | (33.97) | (44.57) | (-7.61) | (44.57) |

(*Continued*)

**Table 12.** (Continued)

| Variable | Path a | Path b | Path c | Path a | Path b | Path c |
|---|---|---|---|---|---|---|
| | *LNAF* | *DA* | *LNAF* | *LNAF* | *DA* | *LNAF* |
| | **OLS Model** | | | **Fixed effects Model** | | |
| | **(1)** | **(2)** | **(3)** | **(4)** | **(5)** | **(6)** |
| *IND* | YES | YES | YES | YES | YES | YES |
| *YEAR* | YES | YES | YES | YES | YES | YES |
| *N* | 31748 | 31748 | 31748 | 31748 | 31748 | 31748 |
| *adj.R²* | 0.6667 | 0.1368 | 0.6669 | 0.6668 | 0.1178 | 0.6669 |

Note (1)

\*\*\*, \*\* and \* indicate significant at 1%, 5% and 10% levels, respectively; (2) Values in parentheses are t-values adjusted for heteroscedasticity.

withdrawal of local government intervention. In the regions with high marketization process, the local government will significantly reduce the political pressure on enterprises, so that enterprises will not bear too many social or political goals, which will reduce the uncertainty of enterprises and further reduce the audit risk. In areas with a low marketization process, the level of economic development is also relatively low. For political purposes such as political performance, local governments are more motivated to intervene in enterprise behavior with their power of resource allocation, which greatly increases the uncertainty of enterprises and the audit risk [81].

In order to verify the heterogeneity of the relationship between employee education level and audit fees in different degrees of marketization process, this paper grouped sample companies by referring to the method of Cheng et al. [82]. Specifically, this paper takes Guangdong, Shanghai, Zhejiang and Jiangsu, the top five provinces in the China Marketization Index: Relative Marketization Process by Region 2011 Report from 2004 to 2009, as the regions with high marketization process. The dummy variable is set as 1, and the other provinces are set as 0. Table 15 shows the group regression results of the low marketization level group and the high marketization level group respectively. In column (1) and (3), the coefficient of employee education level (EDU) is significantly negative at 1% or 10%. In columns (2) and (4), the coefficient of employee education level (EDU) is not significant in the group with high marketization level. The above results show that the effect of employee education level on audit fees mainly exists in the case of low marketization level, and in the case of high marketization level, employee education level has no significant reduction effect on audit fees.

## Confucian culture intensity, staff education level and audit fees

Based on the theory of new institutional economics, in addition to the important role of formal institutions on corporate behavior, informal institutions such as culture also have a significant impact on corporate behavior. Among them, Confucian culture, as the spiritual pillar of the modernization process in China and Southeast Asia, exerts a profound influence on individual decision-making and corporate behavior. To be specific, the Confucian emphasis on "the superior man takes righteousness for profit" and "faithful" work ethic, requirements in the management work of the gentleman to control personal desires [83], thus inhibiting the opportunism behavior of management, alleviate the audit risk caused by information asymmetry [84]; In addition to the Confucian culture "benevolence, righteousness, propriety, wisdom, and trust" thought as the core of traditional Chinese culture, after thousands of years of inheritance and baptism, has become a social moral life and moral behavior norms and constraints of a

**Table 13. Mediation testing based on IC.**

| Variable | Path a | Path b | Path c | Path a | Path b | Path c |
|---|---|---|---|---|---|---|
| | LNAF | IC | LNAF | LNAF | IC | LNAF |
| | OLS Model | | | Fixed effects Model | | |
| | (1) | (2) | (3) | (4) | (5) | (6) |
| EDU | -0.1106*** | -0.1212*** | -0.1035*** | -0.0654** | -0.0700*** | -0.0639** |
| | (-3.07) | (-4.71) | (-2.88) | (-2.19) | (-2.77) | (-2.14) |
| IC | | | 0.0580*** | | | 0.0310*** |
| | | | (5.72) | | | (5.01) |
| SIZE | 0.3777*** | 0.0319*** | 0.3759*** | 0.3015*** | 0.0134*** | 0.3015*** |
| | (47.10) | (6.66) | (46.91) | (42.64) | (2.69) | (42.63) |
| ROA | -0.4745*** | -0.1122 | -0.4680*** | -0.1873*** | -0.0708 | -0.1853*** |
| | (-5.18) | (-1.60) | (-5.12) | (-2.98) | (-1.15) | (-2.95) |
| LEV | 0.0146 | 0.1022*** | 0.0087 | 0.0851** | 0.0686** | 0.0835** |
| | (0.37) | (3.63) | (0.22) | (2.55) | (2.40) | (2.51) |
| LOSS | 0.0292** | 0.0298** | 0.0275* | 0.0198** | 0.0174 | 0.0193** |
| | (1.97) | (2.31) | (1.85) | (2.14) | (1.63) | (2.10) |
| BM | -0.1914*** | -0.0129 | -0.1907*** | 0.0201 | 0.0474** | 0.0182 |
| | (-6.23) | (-0.58) | (-6.22) | (0.98) | (2.34) | (0.88) |
| RIP | -0.0381*** | -0.0099* | -0.0375*** | -0.0260*** | -0.0056 | -0.0259*** |
| | (-4.85) | (-1.69) | (-4.78) | (-4.08) | (-0.96) | (-4.08) |
| QUICK | -0.0132*** | 0.0021 | -0.0133*** | -0.0077*** | 0.0051*** | -0.0079*** |
| | (-4.65) | (1.04) | (-4.71) | (-3.62) | (2.73) | (-3.71) |
| BIG4 | 0.6760*** | 0.0386* | 0.6738*** | 0.2678*** | 0.0388* | 0.2673*** |
| | (17.37) | (1.75) | (17.32) | (8.06) | (1.71) | (8.01) |
| LAGOPINION | -0.1885*** | -0.0280 | -0.1868*** | -0.1048*** | 0.0061 | -0.1053*** |
| | (-8.11) | (-1.64) | (-8.03) | (-5.95) | (0.36) | (-5.98) |
| AUDITTURN | -0.0432*** | 0.0169** | -0.0442*** | -0.0245*** | 0.0084 | -0.0247*** |
| | (-4.76) | (2.08) | (-4.89) | (-4.11) | (1.12) | (-4.15) |
| SOE | -0.0705*** | 0.1046*** | -0.0766*** | -0.0243 | 0.0907*** | -0.0263 |
| | (-4.75) | (9.99) | (-5.14) | (-1.47) | (7.08) | (-1.59) |
| Gender | 0.0277 | -0.0260 | 0.0292 | -0.0154 | -0.0364** | -0.0142 |
| | (1.43) | (-1.64) | (1.51) | (-1.20) | (-2.49) | (-1.10) |
| Age | 0.0010 | 0.0000 | 0.0010 | 0.0008* | -0.0003 | 0.0008* |
| | (1.35) | (0.09) | (1.35) | (1.73) | (-0.53) | (1.76) |
| Share | -0.0172 | -0.0420*** | -0.0147 | -0.0093 | -0.0203*** | -0.0090 |
| | (-1.60) | (-5.26) | (-1.38) | (-1.25) | (-2.75) | (-1.21) |
| Degree | -0.0082 | -0.0725*** | -0.0040 | 0.0091 | -0.0613*** | 0.0108 |
| | (-0.75) | (-8.94) | (-0.37) | (1.29) | (-8.03) | (1.54) |
| Finan | 0.0152 | -0.0296** | 0.0169 | 0.0063 | -0.0422*** | 0.0077 |
| | (0.87) | (-2.17) | (0.97) | (0.59) | (-3.16) | (0.72) |
| Tenure | 0.0037*** | 0.0007 | 0.0037*** | -0.0010 | -0.0003 | -0.0009 |
| | (3.48) | (0.86) | (3.45) | (-1.22) | (-0.35) | (-1.19) |
| Change | -0.0064 | 0.0005 | -0.0064 | -0.0032 | 0.0052 | -0.0034 |
| | (-1.37) | (0.10) | (-1.38) | (-1.07) | (1.28) | (-1.13) |
| StaffDirector | -0.0011 | 0.0315 | -0.0029 | -0.0409 | 0.0542 | -0.0427 |
| | (-0.02) | (0.70) | (-0.06) | (-1.50) | (1.31) | (-1.56) |
| Constant | 5.4397*** | -0.7034*** | 5.4805*** | 6.7484*** | -0.4198*** | 6.7535*** |
| | (33.94) | (-7.20) | (34.26) | (44.57) | (-4.11) | (44.58) |

(*Continued*)

**Table 13.** (Continued)

| Variable | Path a | Path b | Path c | Path a | Path b | Path c |
|---|---|---|---|---|---|---|
| | *LNAF* | *IC* | *LNAF* | *LNAF* | *IC* | *LNAF* |
| | **OLS Model** | | | **Fixed effects Model** | | |
| | **(1)** | **(2)** | **(3)** | **(4)** | **(5)** | **(6)** |
| *IND* | YES | YES | YES | YES | YES | YES |
| *YEAR* | YES | YES | YES | YES | YES | YES |
| *N* | 31748 | 31748 | 31748 | 31748 | 31748 | 31748 |
| *adj.R²* | 0.6667 | 0.1469 | 0.6678 | 0.6668 | 0.1770 | 0.6671 |

Note (1)

\*\*\*, \*\* and \* indicate significant at 1%, 5% and 10% levels, respectively; (2) Values in parentheses are t-values adjusted for heteroscedasticity.

"common law". This kind of informal institution for management decision provides a kind of behavior criterion, reduces the uncertainty of the enterprise, and further reduces the audit risk [85].

In order to verify the heterogeneity of the relationship between employee education level and audit fees when the influence of Confucian culture is different, this paper groups the sample companies by referring to the method of Jin et al. [86]. To be specific, this paper obtains the information of Confucius temples at provincial or municipality level through Baidu, Google search engine and China National Studies Network, and obtains 530 existing Confucian temples whose specific locations can be determined. Then, the number of Confucian temples built in the province where the company is registered is used to measure the influence of Confucian culture of the company. If the number of Confucian temples built in the province where the company is registered is greater than the national median, it is 1; otherwise, it is 0. Table 16 shows the group regression results of the low and high Confucian culture intensity groups respectively. In column (1) and (3), the coefficient of employee education level (EDU) in the low Confucian culture intensity group is significantly negative at the 1% level. In columns (2) and (4), the coefficient of employee education level (EDU) is not significant in the group with high Confucian cultural intensity. The above results show that the influence of employee education level on audit fees mainly exists in the case of low Confucian culture intensity, and in the case of high Confucian culture intensity, employee education level has no significant reduction effect on audit fees.

## Conclusions and implications

The continuous cultivation and optimal allocation of human capital are related to the long-term development of enterprises, the steady growth of economy and the great rejuvenation of the nation. It is a common concern of enterprises, government and academia. Will auditors think that the education level of all employees of enterprises can reduce audit risks by improving the quality of financial statements, so as to reduce audit fees? This paper takes the implementation of the Labor Protection Law in 2008 as an exogenous impact event of the increase in the intensity of labor protection to test whether labor protection can strengthen the inhibitory effect of highly educated employees on audit fees. The results show that highly educated employees can effectively reduce the audit fees borne by the company. However, the implementation of the Labor Protection Law weakens this inhibitory effect, and the above results do not change after a series of stability tests. Further test results show that the inhibitory effect of

**Table 14. Mediation testing based on restatement.**

| Variable | Path a | Path b | Path c | Path a | Path b | Path c |
|---|---|---|---|---|---|---|
| | LNAF | Restatement | LNAF | LNAF | Restatemen | LNAF |
| | OLS Model | | | Fixed effects Model | | |
| | (1) | (2) | (3) | (4) | (5) | (6) |
| EDU | -0.1106*** | -0.0638*** | -0.1093*** | -0.0654** | -0.0606*** | -0.0648** |
| | (-3.07) | (-3.78) | (-3.04) | (-2.19) | (-3.57) | (-2.17) |
| Restatement | | | 0.0189** | | | 0.0107** |
| | | | (2.54) | | | (2.39) |
| SIZE | 0.3777*** | -0.0051* | 0.3778*** | 0.3015*** | -0.0035 | 0.3014*** |
| | (47.10) | (-1.70) | (47.10) | (42.64) | (-1.17) | (42.61) |
| ROA | -0.4745*** | -0.2110*** | -0.4705*** | -0.1873*** | -0.1822*** | -0.1865*** |
| | (-5.18) | (-3.92) | (-5.15) | (-2.98) | (-3.40) | (-2.97) |
| LEV | 0.0146 | 0.0760*** | 0.0132 | 0.0851** | 0.0757*** | 0.0845** |
| | (0.37) | (4.01) | (0.33) | (2.55) | (4.06) | (2.54) |
| LOSS | 0.0292** | 0.0187* | 0.0289* | 0.0198** | 0.0163 | 0.0197** |
| | (1.97) | (1.76) | (1.94) | (2.14) | (1.55) | (2.13) |
| BM | -0.1914*** | -0.0022 | -0.1914*** | 0.0201 | 0.0004 | 0.0200 |
| | (-6.23) | (-0.15) | (-6.24) | (0.98) | (0.03) | (0.98) |
| RIP | -0.0381*** | 0.0019 | -0.0381*** | -0.0260*** | 0.0024 | -0.0260*** |
| | (-4.85) | (0.39) | (-4.86) | (-4.08) | (0.49) | (-4.09) |
| QUICK | -0.0132*** | -0.0017 | -0.0131*** | -0.0077*** | -0.0015 | -0.0077*** |
| | (-4.65) | (-1.16) | (-4.64) | (-3.62) | (-1.05) | (-3.62) |
| BIG4 | 0.6760*** | -0.0568*** | 0.6771*** | 0.2678*** | -0.0573*** | 0.2683*** |
| | (17.37) | (-5.27) | (17.41) | (8.06) | (-5.20) | (8.09) |
| LAGOPINION | -0.1885*** | -0.0536*** | -0.1875*** | -0.1048*** | -0.0447*** | -0.1047*** |
| | (-8.11) | (-3.78) | (-8.07) | (-5.95) | (-3.16) | (-5.94) |
| AUDITTURN | -0.0432*** | 0.0229*** | -0.0437*** | -0.0245*** | 0.0187*** | -0.0246*** |
| | (-4.76) | (3.29) | (-4.81) | (-4.11) | (2.73) | (-4.12) |
| SOE | -0.0705*** | -0.0197*** | -0.0702*** | -0.0243 | -0.0196*** | -0.0241 |
| | (-4.75) | (-3.15) | (-4.72) | (-1.47) | (-3.12) | (-1.46) |
| Gender | 0.0277 | 0.0169 | 0.0274 | -0.0154 | 0.0143 | -0.0154 |
| | (1.43) | (1.62) | (1.42) | (-1.20) | (1.39) | (-1.20) |
| Age | 0.0010 | -0.0004 | 0.0010 | 0.0008* | -0.0003 | 0.0008* |
| | (1.35) | (-0.94) | (1.36) | (1.73) | (-0.72) | (1.73) |
| Share | -0.0172 | -0.0169*** | -0.0168 | -0.0093 | -0.0154*** | -0.0092 |
| | (-1.60) | (-3.09) | (-1.58) | (-1.25) | (-2.88) | (-1.24) |
| Degree | -0.0082 | 0.0093* | -0.0084 | 0.0091 | 0.0084 | 0.0090 |
| | (-0.75) | (1.72) | (-0.76) | (1.29) | (1.56) | (1.28) |
| Finan | 0.0152 | -0.0021 | 0.0153 | 0.0063 | -0.0052 | 0.0065 |
| | (0.87) | (-0.21) | (0.87) | (0.59) | (-0.49) | (0.60) |
| Tenure | 0.0037*** | -0.0005 | 0.0037*** | -0.0010 | -0.0003 | -0.0010 |
| | (3.48) | (-1.00) | (3.49) | (-1.22) | (-0.70) | (-1.22) |
| Change | -0.0064 | -0.0012 | -0.0064 | -0.0032 | -0.0018 | -0.0032 |
| | (-1.37) | (-0.28) | (-1.37) | (-1.07) | (-0.40) | (-1.06) |
| StaffDirector | -0.0011 | 0.0209 | -0.0015 | -0.0409 | 0.0235 | -0.0411 |
| | (-0.02) | (1.04) | (-0.03) | (-1.50) | (1.20) | (-1.51) |
| Constant | 5.4397*** | 0.3324*** | 5.4335*** | 6.7484*** | 0.2788*** | 6.7488*** |
| | (33.94) | (5.05) | (33.89) | (44.57) | (4.28) | (44.57) |

(*Continued*)

**Table 14.** (Continued)

| Variable | Path a | Path b | Path c | Path a | Path b | Path c |
|---|---|---|---|---|---|---|
| | *LNAF* | *Restatement* | *LNAF* | *LNAF* | *Restatemen* | *LNAF* |
| | **OLS Model** | | | **Fixed effects Model** | | |
| | **(1)** | **(2)** | **(3)** | **(4)** | **(5)** | **(6)** |
| *IND* | YES | YES | YES | YES | YES | YES |
| *YEAR* | YES | YES | YES | YES | YES | YES |
| *N* | 31748 | 31748 | 31748 | 31748 | 31748 | 31748 |
| *adj.R²* | 0.6667 | 0.1087 | 0.6668 | 0.6668 | 0.1138 | 0.6668 |

Note (1)

***, ** and * indicate significant at 1%, 5% and 10% levels, respectively; (2) Values in parentheses are t-values adjusted for heteroscedasticity.

**Table 15. Marketization level, employee education level and audit fees.**

| Variable | *LNAF* | | | |
|---|---|---|---|---|
| | **OLS Model** | | **Fixed effects Model** | |
| | **Low level of marketization** | **High level of marketization** | **Low level of marketization** | **High level of marketization** |
| | **(1)** | **(2)** | **(3)** | **(4)** |
| *EDU* | -0.1303*** | 0.0462 | -0.0692* | -0.0012 |
| | (-2.90) | (0.85) | (-1.88) | (-0.02) |
| *SIZE* | 0.3841*** | 0.3607*** | 0.3007*** | 0.2860*** |
| | (36.40) | (32.16) | (34.60) | (28.01) |
| *ROA* | -0.4892*** | -0.4494*** | -0.2212*** | -0.0811 |
| | (-4.30) | (-3.26) | (-2.98) | (-0.84) |
| *LEV* | 0.0299 | 0.0526 | 0.0869** | 0.0981* |
| | (0.60) | (0.82) | (2.06) | (1.89) |
| *LOSS* | 0.0296 | 0.0414* | 0.0072 | 0.0440*** |
| | (1.59) | (1.77) | (0.67) | (2.78) |
| *BM* | -0.1856*** | -0.1754*** | 0.0499* | -0.0006 |
| | (-4.60) | (-4.07) | (1.91) | (-0.02) |
| *RIP* | -0.0331*** | -0.0286** | -0.0176** | -0.0299*** |
| | (-3.36) | (-2.53) | (-2.17) | (-3.12) |
| *QUICK* | -0.0091** | -0.0138*** | -0.0046* | -0.0107*** |
| | (-2.51) | (-3.35) | (-1.66) | (-3.39) |
| *BIG4* | 0.7636*** | 0.5419*** | 0.2807*** | 0.2068*** |
| | (15.01) | (10.25) | (5.63) | (5.73) |
| *LAGOPINION* | -0.1787*** | -0.2308*** | -0.1015*** | -0.1144*** |
| | (-6.66) | (-5.93) | (-4.99) | (-4.07) |
| *AUDITTURN* | -0.0454*** | -0.0340** | -0.0289*** | -0.0189* |
| | (-3.85) | (-2.52) | (-3.90) | (-1.92) |
| *SOE* | -0.0541*** | -0.0123 | -0.0170 | 0.0191 |
| | (-2.84) | (-0.52) | (-0.86) | (0.72) |
| *Gender* | 0.0384 | 0.0060 | -0.0116 | -0.0321* |
| | (1.41) | (0.25) | (-0.72) | (-1.83) |
| *Age* | 0.0014 | 0.0002 | 0.0017** | -0.0001 |

(*Continued*)

**Table 15.** (Continued）

| Variable | LNAF | | | |
|---|---|---|---|---|
| | OLS Model | | Fixed effects Model | |
| | Low level of marketization | High level of marketization | Low level of marketization | High level of marketization |
| | (1) | (2) | (3) | (4) |
| | (1.38) | (0.25) | (2.56) | (-0.16) |
| Share | -0.0283* | -0.0252* | -0.0063 | -0.0120 |
| | (-1.93) | (-1.76) | (-0.70) | (-1.10) |
| Degree | 0.0020 | -0.0110 | 0.0154* | -0.0012 |
| | (0.14) | (-0.72) | (1.69) | (-0.13) |
| Finan | 0.0232 | -0.0128 | 0.0183 | -0.0025 |
| | (0.89) | (-0.58) | (1.22) | (-0.16) |
| Tenure | 0.0034** | 0.0028* | -0.0004 | -0.0009 |
| | (2.35) | (1.89) | (-0.41) | (-0.78) |
| Change | -0.0082 | -0.0001 | -0.0064 | 0.0025 |
| | (-1.30) | (-0.02) | (-1.53) | (0.63) |
| StaffDirector | -0.0418 | 0.0430 | -0.0539 | -0.0246 |
| | (-0.70) | (0.58) | (-1.40) | (-0.68) |
| Constant | 5.2203*** | 6.2291*** | 6.6184*** | 7.6872*** |
| | (24.71) | (21.02) | (37.11) | (19.43) |
| IND | YES | YES | YES | YES |
| YEAR | YES | YES | YES | YES |
| N | 18552 | 13196 | 18552 | 13196 |
| adj.$R^2$ | 0.6868 | 0.6557 | 0.6616 | 0.6416 |

Note (1)

***, ** and * indicate significant at 1%, 5% and 10% levels, respectively; (2) Values in parentheses are t-values adjusted for heteroscedasticity.

highly educated employees on audit fees is more significant when the level of marketization is low and the intensity of Confucian culture is weak.

The results of this paper show that the education level of employees does have a significant impact on audit fees, which indicates that Chinese auditors will consider the impact of the overall education level of employees on the enterprise when making audit pricing. This study confirms that the personal characteristics of employees other than senior executives can influence the auditor's perception and response to corporate risks, which to a certain extent provides motivation and direction for the allocation of corporate human capital, and provides space and support for the auditor's decision optimization. In addition, from the perspective of enterprise employees, this paper directly studies the effect of the implementation of the Labor Protection Law on the individual perspective of employees, which provides a new research perspective for the impact of the implementation of the Labor Protection Law on the level of enterprise spectators. Finally, this paper enriches the theoretical research on corporate governance embedded in Chinese traditional culture, and provides empirical evidence for the governance role of Confucian culture, an informal system. It shows that culture, as an informal institution, can regulate human behavior through the infiltration and assimilation of human consciousness. This conclusion is helpful for enterprises to build a good organizational culture to regulate and guide employee behavior, so as to strengthen human capital.

**Table 16. Confucian cultural intensity, employee education level and audit fees.**

| Variable | LNAF | | | |
|---|---|---|---|---|
| | **OLS Model** | | **Fixed effects Model** | |
| | **Low intensity of Confucian culture** | **High intensity of Confucian culture** | **Low intensity of Confucian culture** | **High intensity of Confucian culture** |
| | **(1)** | **(2)** | **(3)** | **(4)** |
| EDU | -0.1279*** | -0.0443 | -0.1116*** | -0.0230 |
| | (-2.60) | (-0.87) | (-2.75) | (-0.55) |
| SIZE | 0.4003*** | 0.3503*** | 0.3081*** | 0.2759*** |
| | (34.33) | (33.52) | (31.24) | (31.21) |
| ROA | -0.5459*** | -0.3829*** | -0.1720* | -0.2191*** |
| | (-4.04) | (-3.23) | (-1.87) | (-2.84) |
| LEV | -0.0444 | 0.0899 | 0.0585 | 0.1139** |
| | (-0.79) | (1.64) | (1.27) | (2.44) |
| LOSS | 0.0498** | 0.0123 | 0.0329*** | 0.0060 |
| | (2.38) | (0.61) | (2.61) | (0.47) |
| BM | -0.2667*** | -0.1023** | 0.0200 | 0.0484* |
| | (-5.98) | (-2.51) | (0.70) | (1.71) |
| RIP | -0.0293*** | -0.0443*** | -0.0269*** | -0.0197** |
| | (-2.61) | (-4.16) | (-3.13) | (-1.97) |
| QUICK | -0.0135*** | -0.0096** | -0.0079*** | -0.0062** |
| | (-3.47) | (-2.47) | (-2.62) | (-2.18) |
| BIG4 | 0.6955*** | 0.6289*** | 0.2629*** | 0.2423*** |
| | (13.37) | (11.20) | (6.07) | (5.17) |
| LAGOPINION | -0.1930*** | -0.1757*** | -0.1115*** | -0.1135*** |
| | (-6.04) | (-5.36) | (-4.55) | (-5.04) |
| AUDITTURN | -0.0356*** | -0.0494*** | -0.0358*** | -0.0157* |
| | (-2.90) | (-3.67) | (-4.56) | (-1.76) |
| SOE | -0.0469** | -0.0945*** | -0.0146 | -0.0186 |
| | (-2.32) | (-4.46) | (-0.68) | (-0.82) |
| Gender | 0.0087 | 0.0391 | -0.0239 | -0.0074 |
| | (0.32) | (1.48) | (-1.47) | (-0.42) |
| Age | 0.0014 | 0.0005 | 0.0015** | 0.0005 |
| | (1.38) | (0.53) | (2.15) | (0.78) |
| Share | -0.0308** | -0.0034 | -0.0108 | -0.0022 |
| | (-2.01) | (-0.24) | (-1.01) | (-0.24) |
| Degree | -0.0166 | 0.0006 | 0.0063 | 0.0149 |
| | (-1.06) | (0.04) | (0.67) | (1.57) |
| Finan | -0.0137 | 0.0426* | -0.0027 | 0.0202 |
| | (-0.54) | (1.90) | (-0.19) | (1.30) |
| Tenure | 0.0043*** | 0.0036*** | -0.0002 | -0.0009 |
| | (2.72) | (2.67) | (-0.21) | (-0.95) |
| Change | -0.0097 | -0.0011 | -0.0024 | -0.0051 |
| | (-1.47) | (-0.17) | (-0.58) | (-1.23) |
| StaffDirector | 0.0131 | -0.0415 | -0.0548* | -0.0270 |
| | (0.21) | (-0.66) | (-1.82) | (-0.58) |
| Constant | 5.1113*** | 5.8154*** | 6.6438*** | 7.1977*** |
| | (22.39) | (27.45) | (32.79) | (35.59) |
| IND | YES | YES | YES | YES |

*(Continued)*

**Table 16.** (Continued)

| Variable | LNAF | | | |
| --- | --- | --- | --- | --- |
| | OLS Model | | Fixed effects Model | |
| | Low intensity of Confucian culture | High intensity of Confucian culture | Low intensity of Confucian culture | High intensity of Confucian culture |
| | (1) | (2) | (3) | (4) |
| YEAR | YES | YES | YES | YES |
| N | 16400 | 15348 | 16400 | 15348 |
| adj.$R^2$ | 0.6906 | 0.6406 | 0.6573 | 0.6467 |

Note (1)

***, ** and * indicate significant at 1%, 5% and 10% levels, respectively; (2) Values in parentheses are t-values adjusted for heteroscedasticity.

## Supporting information

**S1 Appendix. The process of enacting and amending the labor protection law in China.**
(DOCX)

**S1 Fig. Employee education level.**
(PDF)

## Author Contributions

**Writing – original draft:** Xiaotian Shen, Anni Wu, Yi Ding, Qian Sun.

**Writing – review & editing:** Mengge Liu.

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
