## [Decision Letter · Decision Letter 0]

27 Nov 2023

PONE-D-23-32966Employee Education, Labor Protection Intensity and Auditor Risk PerceptionPLOS ONE

Dear Dr. Liu,

Thank you for submitting your manuscript to PLOS ONE. After careful consideration, we feel that it has merit but does not fully meet PLOS ONE’s publication criteria as it currently stands. Therefore, we invite you to submit a revised version of the manuscript that addresses the points raised during the review process.

We look forward to receiving your revised manuscript.

Kind regards,

Rana Muhammad Ammar Zahid, PhD

Academic Editor

PLOS ONE

Additional Editor Comments:

Dear Authors,

After careful consideration and review, we request major amendments to this manuscript before accepting it for publication.

Please answer the reviewers comments to improve the manuscript, as it has area to improve in all section, introduction, hypothesis development, methodology and conclusion. Here are few tips and some recommended paper that will help you to revise the manuscript.

Please address the issues raised by the reviewers regarding the theoretical part of hypothesis development. Moreover, there is an important strand of literature with respect to the Audit Quality in following studies. These studies will help you to improve the methodology as well.

ESG, Dividend Payout Policy and Audit Quality: An empirical Evidence from the Western Europe

Borsa Istanbul Review

https://doi.org/10.1016/j.bir.2022.10.012

The role of Audit Quality in ESG-Corporate Financial Performance nexus: empirical evidence from western European companies

Borsa Istanbul Review

https://doi.org/10.1016/j.bir.2022.08.011

Moderating role of audit quality in ESG performance and capital financing dynamics: insights in China. Environ Dev Sustain (2023). https://doi.org/10.1007/s10668-023-03636-9

Make sure to proofread the manuscript before it is resubmitted to the journal. Please go through the journal’s guidelines thoroughly and revise the paper accordingly. Thank you for submitting your paper to the PlosOne.

Reviewers' comments:

Reviewer's Responses to Questions

**Comments to the Author**

1. Is the manuscript technically sound, and do the data support the conclusions?

Reviewer #1: Yes

Reviewer #2: Yes

2. Has the statistical analysis been performed appropriately and rigorously? 

Reviewer #1: Yes

Reviewer #2: Yes

3. Have the authors made all data underlying the findings in their manuscript fully available?

Reviewer #1: Yes

Reviewer #2: Yes

4. Is the manuscript presented in an intelligible fashion and written in standard English?

Reviewer #1: Yes

Reviewer #2: Yes

5. Review Comments to the Author

Reviewer #1: I was pleased to read the article Employee Education, Labor Protection Intensity and Auditor Risk Perception. This study is needed. However, there are a number of issues that need to be addressed.

An abstract should describe the purpose or need of the study.

In my opinion, the introduction section was very lengthy. This section explains the study's context and objective. The research gap should also be narrowed after analyzing previous studies. The research method is not adequately explained in the first section.

The results of this study are not supported by significant and recent literature, although they are relevant and have been adequately discussed. Additionally, the most recent research articles should be cited. Please refer to https://doi.org/10.1016/j.net.2022.05.022; as https://doi.org/10.1177/21582440221116113

Reviewer #2: Comments:

1. Authors claim that hand-collected data of employees' education level. Why did this happen? Is important to figure out that this manuscript is submitted to an international journal and different readers of different countries have different knowledge. E.g., in the EU, information about employees' education degree is mandatory; every year firms must release such information, which is compiled in an official database. It would be interesting to know if the authors did not have access to such dataset, or if simply it does not exist in China.

2. Once, and using the same arguments as before, It would be interesting to learn more about the passage of the Labor Protection Law in China. Might be as an appendix (tip: search in papers about the effects of EU directives enactment).

3. Lines 56-61. This sentence must be reformulated because ideas are too messy: “At present, existing studies (…)

4. Lines 174-175. The authors cite Dechow and Dichev (2002) to support the following idea: “Previous studies have shown that in terms of mandatory information disclosure, higher employee education (…)”; the work of Dechow and Dichev (2002) does not perfectly fit in the spirit of such statement. I suggest finding a different study to support it.

5. I suggest simplifying hypothesis 1. E:g. “Employees with high educational background can effectively reduce the audit fees borne by the company.”

6. In the “Descriptive statistics” section, avoid the expression “certain”. It is not accurate and does not match the scientific method spirit. Perhaps it could be replaced by “suggests”.

7. In Table 4, variables LNAF and SIZE should be presented in monetary units.

8. Also find another expression to “Hypothesis (1) 2 has been verified”; e.g., “evidence supports Hypothesis (1) 2”.

9. At this point, I’m wondering if auditor rotation is mandatory in China. I suggest some clarification about that issue.

10. And about the employee director system in China, is employee-board representation mandatory? Do you anticipate any impact on controlling for that in your current results?

11. The section “Sensitivity test of estimation method” could be included as a footnote because it does not really add anything new.

12. The “Mediation effect test” section is confusing because the number of Tables is wrong (see Minor Errors) and repeating the same equations is confusing. I suggest elaborating more on the rationale behind this analysis.

13. A gap in this study is the absence of coefficients’ estimates interpretation. For example, the interaction coefficients in Table 6 should be interpreted. If one sums EDU+ LAW+ EDU×LAW, one will find a positive value, meaning the passage of the Law surpasses the EDU effect.

Minor Issues:

- Revise the number of Tables. E.g: line 353 refers to Table 4, but the paragraph reports to Table 6.

- Acronyms such as CSMAR, ST and *ST should be described.

- Lines 80/81: In this sentence: “Past research mainly focuse on the audit background of the executive (…)” the right form is “focuses”.

- References 36 and 37 are repeated.

- Line 190: Spells “whistleblowing”.

- Line 235: In this sentence: “Based on this, this paper puts forward the following hypotheses”, is not hypotheses but hypothesis.

6. PLOS authors have the option to publish the peer review history of their article (what does this mean?). If published, this will include your full peer review and any attached files.

Reviewer #1: **Yes: **Dr. Shahid Ali

Reviewer #2: No

---

## [Author Response · Author response to Decision Letter 0]

9 Jan 2024

Manuscript Title: Employee Education, Labor Protection Intensity and Auditor Risk Perception

Manuscript ID: PONE-D-23-32966

Response to Editor and Reviewer

Dear Editor and Reviewer,

We are most grateful to you for giving us another opportunity to revise and resubmit our paper. We would like to express our deep gratitude towards you for the clear, constructive, and insightful comments that have helped us further improve the manuscript. We took full account of all points that you raised and have revised the paper accordingly. We believe that the quality of the manuscript, as a whole, has been greatly improved by these revisions, which we hope will resolve and address your concerns.

Below we explain how we respond to your comments, detailing our revisions and clarifications. To ensure that the comments and our responses are clearly identified, the former are shown in italicized blue text, and our response to each comment is given underneath the queries in black. Please find our specific reply in the attachment.

We thank you so much for your comments on our work. Your suggestions were vital to further enhance the development of our paper and improve its overall contribution. We have, as explained below, carefully and thoughtfully addressed your concerns in this revised manuscript.

With all best wishes,

Authors

---

## [Decision Letter · Decision Letter 1]

29 Jan 2024

PONE-D-23-32966R1Employee Education, Labor Protection Intensity and Auditor Risk PerceptionPLOS ONE

Dear Dr. Liu,

Thank you for submitting your manuscript to PLOS ONE. After careful consideration, we feel that it has merit but does not fully meet PLOS ONE’s publication criteria as it currently stands. Therefore, we invite you to submit a revised version of the manuscript that addresses the points raised during the review process.

We look forward to receiving your revised manuscript.

Kind regards,

Rana Muhammad Ammar Zahid, PhD

Academic Editor

PLOS ONE

Journal Requirements:

**Additional Editor Comments:**

Dear Authors,

After careful consideration and review, we request minor amendments to this manuscript before accepting it for publication.

Make sure to proofread the manuscript before it is resubmitted to the journal. Please go through the journal’s guidelines thoroughly and revise the paper accordingly. Thank you for submitting your paper to the PlosOne.

Reviewers' comments:

Reviewer's Responses to Questions

**Comments to the Author**

1. If the authors have adequately addressed your comments raised in a previous round of review and you feel that this manuscript is now acceptable for publication, you may indicate that here to bypass the “Comments to the Author” section, enter your conflict of interest statement in the “Confidential to Editor” section, and submit your "Accept" recommendation.

Reviewer #2: All comments have been addressed

2. Is the manuscript technically sound, and do the data support the conclusions?

Reviewer #2: Yes

3. Has the statistical analysis been performed appropriately and rigorously? 

Reviewer #2: Yes

4. Have the authors made all data underlying the findings in their manuscript fully available?

Reviewer #2: Yes

5. Is the manuscript presented in an intelligible fashion and written in standard English?

Reviewer #2: Yes

6. Review Comments to the Author

Reviewer #2: Minor Issues:

- Include a footnote describing succinctly the auditor rotation process in China

- Revise the number of Tables. E.g: Revise lines 257/258

- Line 216: In this sentence: “Based on this, this paper puts forward the following hypotheses”, replace hypotheses per hypothesis.

7. PLOS authors have the option to publish the peer review history of their article (what does this mean?). If published, this will include your full peer review and any attached files.

Reviewer #2: No

---

## [Author Response · Author response to Decision Letter 1]

30 Jan 2024

Manuscript Title: Employee Education, Labor Protection Intensity and Auditor Risk Perception

Manuscript ID: PONE-D-23-32966R1

Response to Editor and Reviewer 

Dear Editor and Reviewer,

We are most grateful to you for giving us another opportunity to revise and resubmit our paper. We would like to express our deep gratitude towards you for the clear, constructive, and insightful comments that have helped us further improve the manuscript. We took full account of all points that you raised and have revised the paper accordingly. We believe that the quality of the manuscript has been greatly improved by these revisions, which we hope will resolve and address your concerns.

Below we explain how we respond to your comments, detailing our revisions and clarifications. To ensure that the comments and our responses are clearly identified, the former are shown in italicised blue text, and our response to each comment is given underneath the queries in black.

With all best wishes,

Authors

1. Review Comments to the Author

Reviewer #2: Minor Issues:

- Include a footnote describing succinctly the auditor rotation process in China

- Revise the number of Tables. E.g: Revise lines 257/258

- Line 216: In this sentence: “Based on this, this paper puts forward the following hypotheses”, replace hypotheses per hypothesis.

Responses:

We thank you very much for your constructive comments and suggestions. We are sorry that we have neglected some details, and we have made modifications according to your suggestions.

---

## [Editor Report · Decision Letter 2]

2 Feb 2024

Employee Education, Labor Protection Intensity and Auditor Risk Perception

PONE-D-23-32966R2

Dear Dr. Liu,

We’re pleased to inform you that your manuscript has been judged scientifically suitable for publication and will be formally accepted for publication once it meets all outstanding technical requirements.

Kind regards,

Rana Muhammad Ammar Zahid, PhD

Academic Editor

PLOS ONE

Additional Editor Comments (optional):

Thank you for incorporating suggested changes.
---

## [Editor Report · Acceptance letter]

29 Apr 2024

PONE-D-23-32966R2 

PLOS ONE

Dear Dr. Liu, 

I'm pleased to inform you that your manuscript has been deemed suitable for publication in PLOS ONE. Congratulations! Your manuscript is now being handed over to our production team.

Kind regards, 

on behalf of

Dr. Rana Muhammad Ammar Zahid 

Academic Editor

PLOS ONE